# The specific features of the developing T cell compartment of the neonatal lung are a determinant of respiratory syncytial virus immunopathogenesis

Thomas Démoulins[1,2], Melanie Brügger[1,2,3], Beatrice Zumkehr[1,2], Blandina I. Oliveira Esteves[1,2], Kemal Mehinagic[1,2,3,4], Amal Fahmi[1,2,3], Loïc Borcard[1,2,3], Adriano Taddeo[1,2], Damian Jandrasits[1,2], Horst Posthaus[4,5], Charaf Benarafa[1,2], Nicolas Ruggli[1,2], Marco P. Alves[1,2]*

1 Institute of Virology and Immunology, Bern, Switzerland, 2 Department of Infectious Diseases and Pathobiology, Vetsuisse Faculty, University of Bern, Bern, Switzerland, 3 Graduate School for Cellular and Biomedical Sciences, University of Bern, Bern, Switzerland, 4 Institute of Animal Pathology, Department of Infectious Diseases and Pathobiology, Vetsuisse Faculty, University of Bern, Bern, Switzerland, 5 COMPATH, Vetsuisse Faculty & Faculty of Medicine, University of Bern, Bern, Switzerland

* marco.alves@vetsuisse.unibe.ch

## Abstract

The human respiratory syncytial virus (RSV) is a major cause of severe lower respiratory tract infections in infants, possibly due to the properties of the immature neonatal pulmonary immune system. Using the newborn lamb, a classical model of human lung development and a translational model of RSV infection, we aimed to explore the role of cell-mediated immunity in RSV disease during early life. Remarkably, in healthy conditions, the developing T cell compartment of the neonatal lung showed major differences to that seen in the mature adult lung. The most striking observation being a high baseline frequency of bronchoalveolar IL-4-producing CD4+ and CD8+ T cells, which declined progressively over developmental age. RSV infection exacerbated this pro-type 2 environment in the bronchoalveolar space, rather than inducing a type 2 response *per se*. Moreover, regulatory T cell suppressive functions occurred very early to dampen this pro-type 2 environment, rather than shutting them down afterwards, while γδ T cells dropped and failed to produce IL-17. Importantly, RSV disease severity was related to the magnitude of those unconventional bronchoalveolar T cell responses. These findings provide novel insights in the mechanisms of RSV immunopathogenesis in early life, and constitute a major step for the understanding of RSV disease severity.

## Author summary

By using a translational model with full accessibility to the small airways at defined early life periods, we provide a characterization of the developing T cell compartment in the distal lungs of healthy and RSV-infected neonates. This process is highly dynamic and

**Data Availability Statement:** All relevant data are within the manuscript and its Supporting Information files.

**Funding:** T.D. was supported in part by the EU FP7 Project UniVax (HEALTH-F3-2013-60173). This project was funded by the Swiss National Science Foundation (grant No. 310030_172895 to M.P.A.). The funders had no role in study design, data collection and analysis, decision to publish, or preparation of the manuscript.

**Competing interests:** The authors have declared that no competing interests exist.

tightly regulated, characterized by colonizing T cell subsets that synergize towards a narrow pro-tolerogenic immunological window. We believe our work constitutes a solid basis to clarify the age dependency of RSV immunopathogenesis, and should be considered in vaccine design, which remains challenging after five decades of effort.

## Introduction

The human respiratory syncytial virus (RSV) is a seasonal virus, known as the major cause of lower respiratory tract infection during early childhood [1,2]. Worldwide, RSV leads to around 160'000 deaths each year [3]. Although there are risk factors for RSV severity such as prematurity and congenital heart disease, the majority of hospitalized infants are previously healthy [4–7], suggesting that the disease is partially linked to inherent properties of early life immunity. The precise understanding of the immune-driven susceptibility of neonates is essential, considering there is no licensed vaccine and available prophylactic treatments are mostly based on palivizumab, a neutralizing antibody used in high-risk group infants [8].

The first exposure to pathogens occurs during the early postnatal period and is critical for lung colonization by immune cells. There is increasing evidence that the overall neonatal T cell compartment is distinctive during this temporal window [9]. Upon stimulation, neonates have a higher frequency of T cells that differentiate into regulatory T cells (Tregs) compared to adults, to facilitate self-tolerance to developing organs [10]. Also, the developing lung is characterized by a type 2 immune phenotype, and the pulmonary T cell responses present a T helper type 2 (Th2) bias [11,12]. Remarkably, neonatal γδ T cells, a T cell subset enriched at mucosal barrier sites, have an impaired capacity to respond to stimulation [13–16].

Due to the aforementioned reasons, young infants do not mount a classical pulmonary immune response to pathogens such as RSV, which may contribute to high morbidity and mortality [17,18]. A likely explanation could be a combined contribution of viral load [19] and an inappropriate and/or dysfunctional immune response, through an over-exuberant inflammatory response or a biased T cell response [9,20,21]. Neonatal T cells may have both protective and harmful effects in responses to RSV infection. While children with T cell deficiencies cannot efficiently clear the virus [22,23], it remains controversial whether the severity of RSV disease is associated with a bias towards type 2 cytokines such as IL-4 [24–26].

Knowledge of the course of RSV disease is still limited due to ethical and technical difficulties associated with the investigation of the immune response in human neonates. The neonatal lamb model of human RSV infection recapitulates most of the features of the paediatric disease [27]. Furthermore, due to similarities in development and pulmonary immune system, the ovine lung is a classical model of the human respiratory tract [28]. Human RSV-infected animals clinically present rhinitis, coughing and respiratory distress. Typical pulmonary lesions are bronchiolitis and interstitial pneumonia associated with inflammatory infiltrates [27,29,30]. Herein, this model, immunologically related to human, was exploited to have full access to the small airways and follow the development of the pulmonary immune cell features in early life. Moreover, by peribronchial lymph nodes (LNs), lung, and bronchoalveolar space sampling following RSV infection, we assessed the potential contributors of the host neonatal T cell compartment (i.e., CD4+ and CD8+ T cells, γδ T cells and Tregs) over the course of RSV disease.

## Results

### RSV A2 infection is causing lower respiratory tract disease and leading to a potentially protective humoral response

Notably, RSV A2 proved to replicate in ovine well-differentiated airway epithelial cell (WD-AEC) cultures, with a peak of infectious release at 72–96 h, as shown in S1A Fig. Next, to monitor the immune responses during the acute, recovery and convalescence phases of RSV disease, time-points were selected as follow: 3 and 6 days post-infection (p.i.), to evaluate the innate and early adaptive responses; 14 days p.i. to assess the adaptive mechanisms and virus clearance; 42 days p.i. to investigate the impact of RSV A2 infection on the immune cell colonization of the developing lung. The developmental age of sheep does not follow a linear curve compared to humans but in terms of lung development, animals euthanized at day 3, 6, and 14

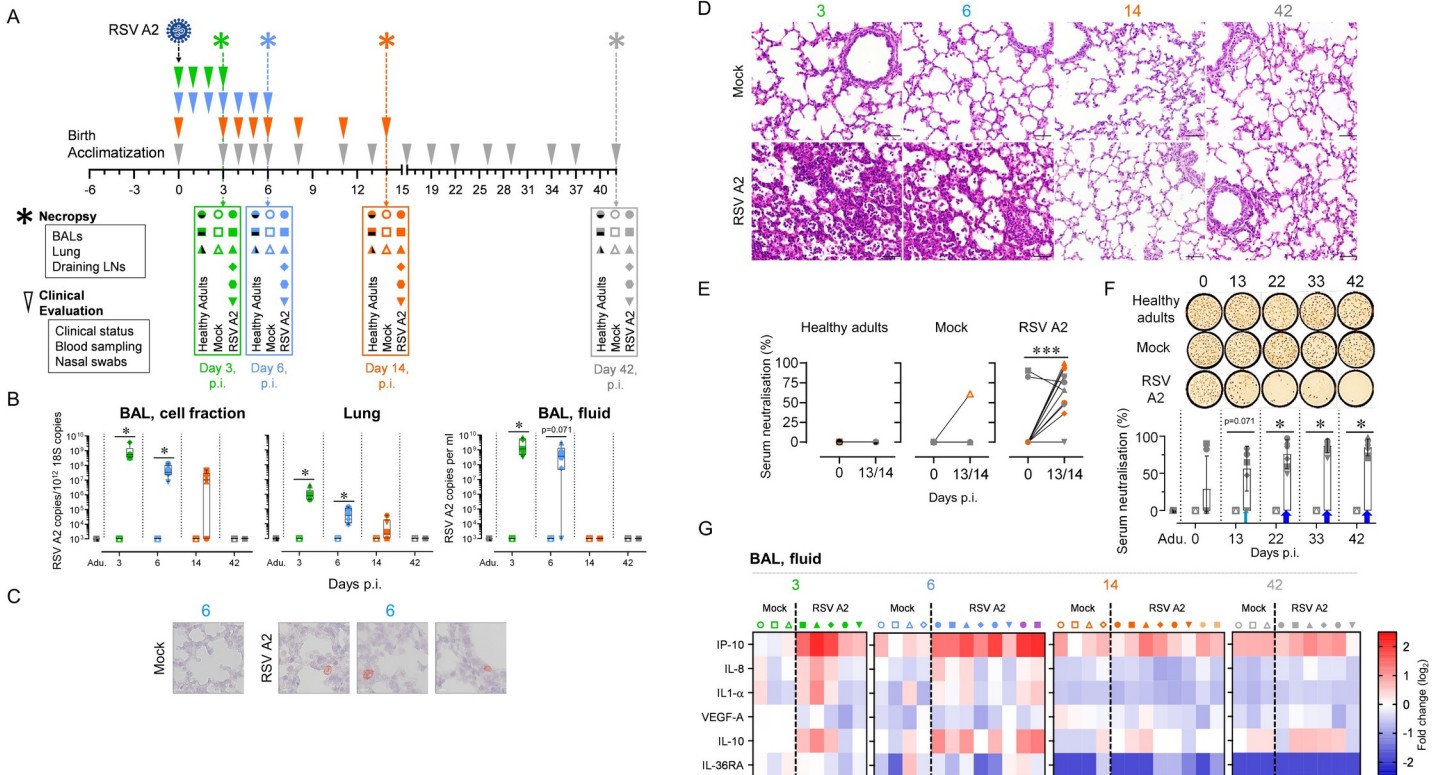

**Fig 1. RSV A2 infection is causing lower respiratory tract disease and leading to a potentially protective humoral response.** (A) Illustration of the trans-tracheal RSV A2 infection experimental design. RSV A2-infected animals were injected with $10^8$ PFU of RSV A2. As controls, PBS-injected newborns and non-infected adults were used. Before and after RSV A2 infection, clinical evaluation was assessed daily at early time points (between day 0 and day 6) and twice a week at later time points (between day 7 and day 42). Necropsies were done at days 3, 6, and 14 or 42 p.i. (B) RSV A2 was quantified in the respiratory tract (BAL, cell fraction; Lung; BAL, fluid) by qPCR. (C) RSV A2-positive cells detection by immunohistochemistry in lung tissue section from RSV A2-infected neonate (Mock = mock neonate). (D) Representative histological lung tissue sections from neonates over the course of RSV A2 infection (40X, HE). Lungs of RSV A2-infected animals show interstitial septae expanded by inflammatory cells (interstitial pneumonia) and accumulation of inflammatory cells within alveoli at days 3 and 6 p.i. No lesions at days 14 and 42 p.i. The lungs of mock neonates allow comparison with normal lung architecture within the first weeks of life. (E) Neonates infected with RSV A2 can mount an antiviral neutralizing-antibody response. Serums collected in animals prior RSV A2 infection (0), and at days 13 or 14 p.i. (13/14), were co-incubated with RSV A2 (100 PFU) and applied to HEp-2 cells for 48 hours. (F) Anti-RSV A2 Nabs persist over a 42-day long period. Serums collected in animals prior RSV A2 infection (0), and at days 13, 22, 33, 42 p.i. were co-incubated with RSV A2 (100 PFU) and applied to HEp-2 cells for 48 hours. (G) Multiplex immunoassay using BAL from all neonates included in the study. Cytokine concentrations were quantified with a multiplex assay and a single measurement was done per animal tested. Heat map showing $\log_2$-fold changes in concentration of 6 measured cytokines/chemokines normalized by mean value of control group: Mock, 3 days p.i. (A-G) Each symbol represents an individual animal (healthy adults, n = 12; mock neonates, n = 3 per time point; neonates infected with RSV, n = 6 per time point). Boxplots indicate median value (center line) and interquartile ranges (box edges), with whiskers extending to the lowest and the highest values. Adu., healthy adults. Groups were compared using Mann–Whitney U tests (B, F) and Wilcoxon tests (E). Stars indicate significance levels. *, $p < 0.05$; ***, $p < 0.001$.

post-infection (p.i.) can be considered as infants and at day 42 as adolescents [31]. As controls, non-infected newborns and adults were used (Fig 1A).

RSV A2 loads were followed daily using nasal swabs on a 6 day-duration period. Shortly after inoculation (day 1–2 p.i., depending on the animal), a peak followed by a rapid decline of virus load was observed; this first peak was due to a caught reflex triggered by the intra-tracheal injection. Remarkably, RSV loads subsequently re-increased to achieve a second peak in 5 out of 6 animals (day 2–5 p.i., depending on the animal), indicating the capacity of RSV A2 to replicate in our experimental model (S1B Fig). We quantified RSV in BALs (cellular and fluid fractions) and in the lung tissue. Although RSV clearance was slower in the cellular fraction of BALs, it became undetectable in any sample at day 42 p.i., demonstrating an efficient clearance (Fig 1B). Immunohistochemistry of lung tissue sections revealed the presence of RSV-antigens at day 6 p.i. in few cells, presumably macrophages (Fig 1C). Clinically, RSV infected neonates showed signs of rhinitis with clear nasal discharge, occasional coughing and wheezing. Macroscopical lesions were visible at days 3–6 p.i, including failure of pulmonary collapse and focal, dark-red areas of subpleural pulmonary consolidation and atelectasis. Histological lesions, present at days 3–6 p.i consisted of multifocal-coalescing areas of lymphocytic and histiocytic infiltrates within interalveolar septae (interstitial pneumonia) and accumulation of inflammatory cells within alveolar spaces and bronchioles (bronchiolitis) (Fig 1D). Longitudinal measurements of white blood counts and neutrophil counts in the circulation showed no difference between mock and infected neonates (S2 Fig). Next, we monitored the presence of RSV neutralizing antibodies (NAbs) in the serum of all animals before initiating the trial (animal groups euthanized at days 3 and 6: S3 Fig; animal groups euthanized at days 14 and 42: S4 Fig). Besides a sibling (see below), none of them showed neutralizing activity. Then, in the serum of healthy adults and mock neonates, no RSV NAbs induction were detected up to 13–14 days p.i. (Figs 1E and S4). In contrast, RSV A2-infected neonates have naturally acquired NAbs at day 13–14 p.i., except one animal. Notably, two siblings had RSV NAbs preceding infection; this was passively transmitted by their mother whose serum also had a strong RSV-neutralization (S5 Fig). Whether these NAbs were induced following infection by bovine RSV and cross-reacted with human RSV or the result of a random V-D-J recombination that specifically recognizes human RSV epitopes, was out of the scope of the present study. We extended the assay to later time points to test the persistence of RSV neutralization. No loss of humoral memory was measured up to 42 days p.i. (Fig 1F). Altogether, these results indicate that neonates can mount an efficient humoral response suggesting potential protection for a subsequent RSV infection. When quantifying the cytokine profiles in BALs from mock control and RSV-infected animals, the most noticeable induction upon infection were IL-10 (day 6 p.i.), IL-8 and CXCL10/IP-10 (days 3 and 6 p.i.). The latter being a potent lymphocyte chemoattractant (Fig 1G).

## Neonatal RSV A2 infection induces bronchoalveolar space recruitment of activated pDCs

Neonatal RSV infection did not significantly alter the absolute counts in BALs when compared to relative mock neonates, besides a trend to increase at day 14 p.i. (S6A Fig, p = 0.095). We first assessed central components of the pulmonary immune responses bridging innate and adaptive responses, namely plasmacytoid DCs (pDCs). In RSV disease, pDCs are known to be involved during the early stages of the pulmonary immune response and have a protective function by modulating the local immune microenvironment [32]. When quantified in the BALs, pDC numbers of mock neonates were similar to those of healthy adults. Upon RSV infection, we observed a trend to pDC recruitment as early as 3 days p.i., and this was

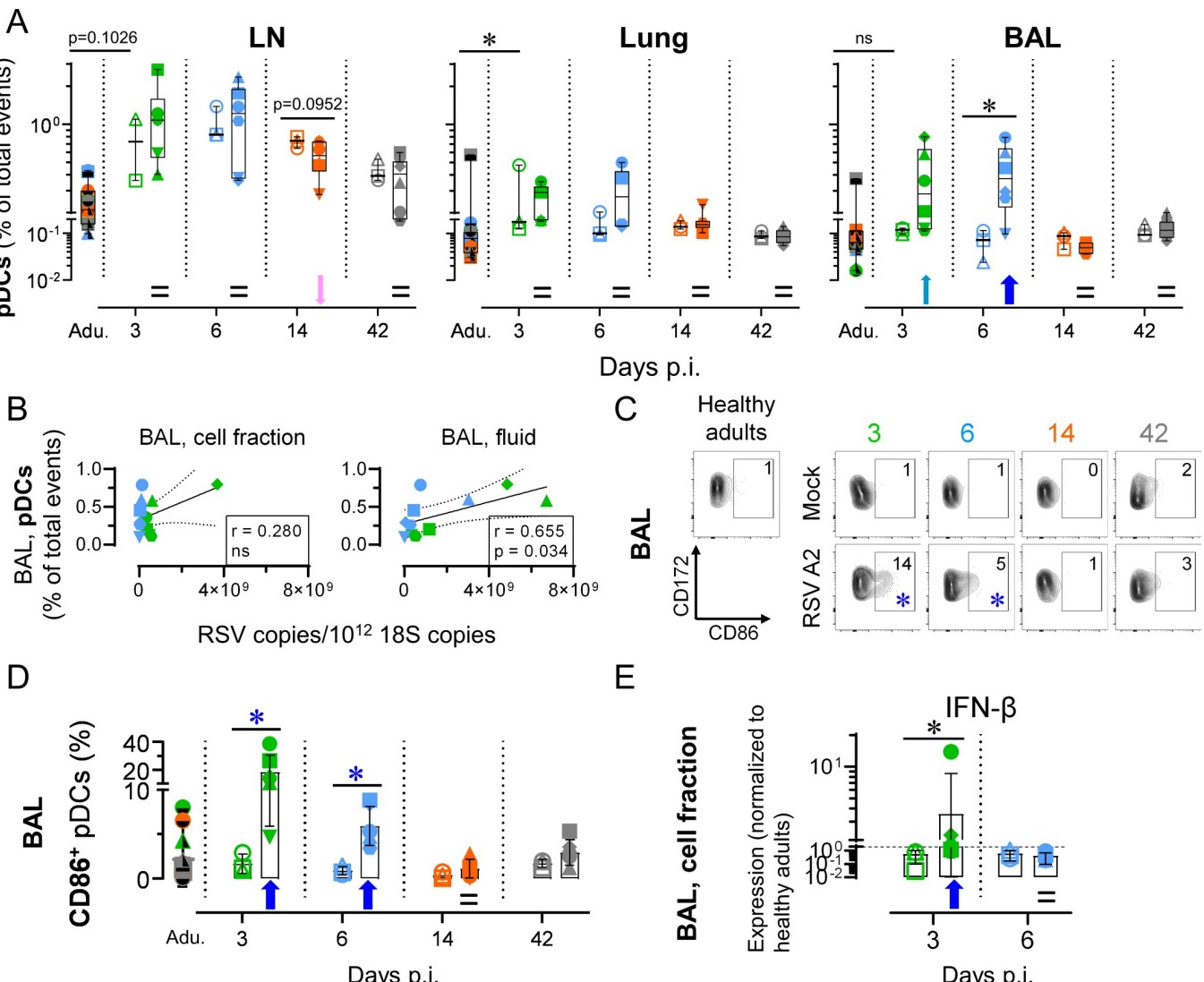

**Fig 2. Neonatal RSV A2 infection induces bronchoalveolar space recruitment of activated pDCs.** (A) Neonatal RSV A2 infection induces an early and transient bronchoalveolar space colonization by pDCs, but not in peribronchial LNs or lung tissue. (B) Correlation coefficient (r) obtained with pDC counts calculated as a function of RSV A2 copies/$10^{12}$ 18S copies. Significance was reached for BAL fluid but not for BAL cellular fraction. (C-D) A high frequency of recruited pDCs in bronchoalveolar space upregulate CD86. (C) Representative FCM contour plots. Numbers indicate the mean percentage of animals per group per time point. Blue asterisks indicate significant differences between RSV A2 and Mock groups for a given time point. (D) As in (C), but with plot displaying all individuals. (E) Type I IFN quantification by qPCR in the BALs of RSV A2 infected neonates and healthy controls. (A-E) Each symbol represents an individual animal (healthy adults, n = 12; mock neonates, n = 3 per time point; neonates infected with RSV, n = 6 per time point). Boxplots indicate median value (center line) and interquartile ranges (box edges), with whiskers extending to the lowest and the highest values. Adu., healthy adults. Groups were compared using Mann–Whitney U-tests (A, C-E). Stars indicate significance levels. *, p < 0.05.

enhanced at day 6 p.i. (p < 0.05). This recruitment was specific of bronchoalveolar space, since we failed to detect any increase in peribronchial LN and lung tissue. However, the pDC recruitment in the bronchoalveolar space was transitional and tightly regulated, since day 14 p. i. corresponded to a return to steady-state levels (gating strategy in S7A Fig, % of total events in Fig 2A, absolute count in S6B Fig). To verify whether the extent of pDC recruitment was linked to the magnitude of RSV shedding, pDC counts were plotted as a function of RSV copies in the fluid fraction of BALs. A significant association was found (Fig 2B). We next evaluated the maturation and activation phenotype of recruited pDCs. A significant CD86

upregulation was found at days 3–6 p.i. (Fig 2C and 2D). This upregulation was transitional and back to steady-state levels at day 14 p.i. In line with the early pDC recruitment and activation, we observed a significant upregulation of IFN-β during the acute phase of neonatal RSV infection, suggesting the induction of an antiviral state in the respiratory tract (Fig 2E). It should be noted that no Ab for ovine type I IFN detection are available, explaining why we employed instead mRNA expression levels.

## Neonatal RSV A2 infection leads to a transient depletion of γδ T cells and a deficiency of IL-17-mediated immunity

Similar to pDCs, γδ T cells link the innate and adaptive arms of the immune system, particularly at mucosal sites including the respiratory tract. Thus, we wondered whether they were involved in the early immune response to RSV infection [33]. γδ T cell numbers in the peribronchial LNs, lung tissue and BALs of mock neonates were comparable to those calculated in healthy adults. However, neonatal RSV infection led to a noticeable decrease in γδ T cell frequency in the BALs at day 14 p.i., whereas this remained stable in other lung compartments. Nevertheless, this γδ T cell depletion was entirely reversed at day 42 p.i. (gating strategy in S7B Fig, % of total events in Fig 3A, absolute counts in S6C Fig). To exclude the possibility that the γδ T cell reduction could be due to migration or relocation, we calculated their counts as a function of the percentage of Live/Dead-positive cells. A negative association was found, suggesting that neonatal γδ T cells death is linked to the presence of RSV in the bronchoalveolar space (Fig 3B). Importantly, the decrease of γδ T cells was not the indirect consequence of the relative increase of another cell subset, since no negative correlation was found when the % of total events of γδ T cells was plotted with % of total events of pDCs, CD4$^+$ T cells, CD8$^+$ T cells and Tregs (S6G Fig). We next ascertained whether neonatal γδ T cells in BALs remained immature during the recovery phase. Remarkably, in healthy adults, half of the γδ T cell pool was composed of mature CD25$^+$ cells, whereas the latter were absent in mock neonates at day 3. The proportion of mature γδ T cells progressively strengthened over time, without reaching the level detected in healthy adults (Fig 3C and 3D). RSV infection exerts an effect on this maturation of neonatal γδ T cells, by depleting significantly the immature WC1$^+$CD25$^-$ pool. However, the observed depletion was followed by a complete replenishment at day 42 p.i., comparable to mock neonates (Fig 3E). Finally, we demonstrated the inability of γδ T cells to produce detectable levels of IL-17 during the neonatal period (days 3–14 p.i.; Fig 3F and 3G).

## Neonatal RSV A2 infection induces a fast Tc2 influx in the bronchoalveolar space

Assessment moved on to adaptive immune response, and more particularly to IL-4 *versus* IFN-γ production by CD4$^+$ or CD8$^+$ T cells, a determining factor in the outcome of protective *versus* harmful immune response to RSV infection [25,34] (gating strategy in S7C Fig; % of total events in Fig 4A and 4B; absolute counts in S6D and S6E Fig). In BALs, mock neonates had a very low number of CD4$^+$ and CD8$^+$ T cells compared to healthy adults, whereas numbers were comparable in draining LNs. RSV infection led to their strong recruitment in the bronchoalveolar space, albeit with distinct kinetics. An unexpected reduction of CD4$^+$ T cell recruitment was seen as soon as 14 days p.i., although not significant (% of total event at day 14 p.i. versus day 6 p.i.: p = 0.0649). This contradicted the values obtained for CD8$^+$ T cells, where the significant enhancement following RSV infection was maintained. Interestingly, this influx of CD8$^+$ T cells was preceded by their significant increase in the lung, indicating that they transit through this compartment. When we considered both CD4$^+$ and CD8$^+$ fractions at a later time-point, the numbers were comparable to those of mock neonates (Figs 4A, 4B,

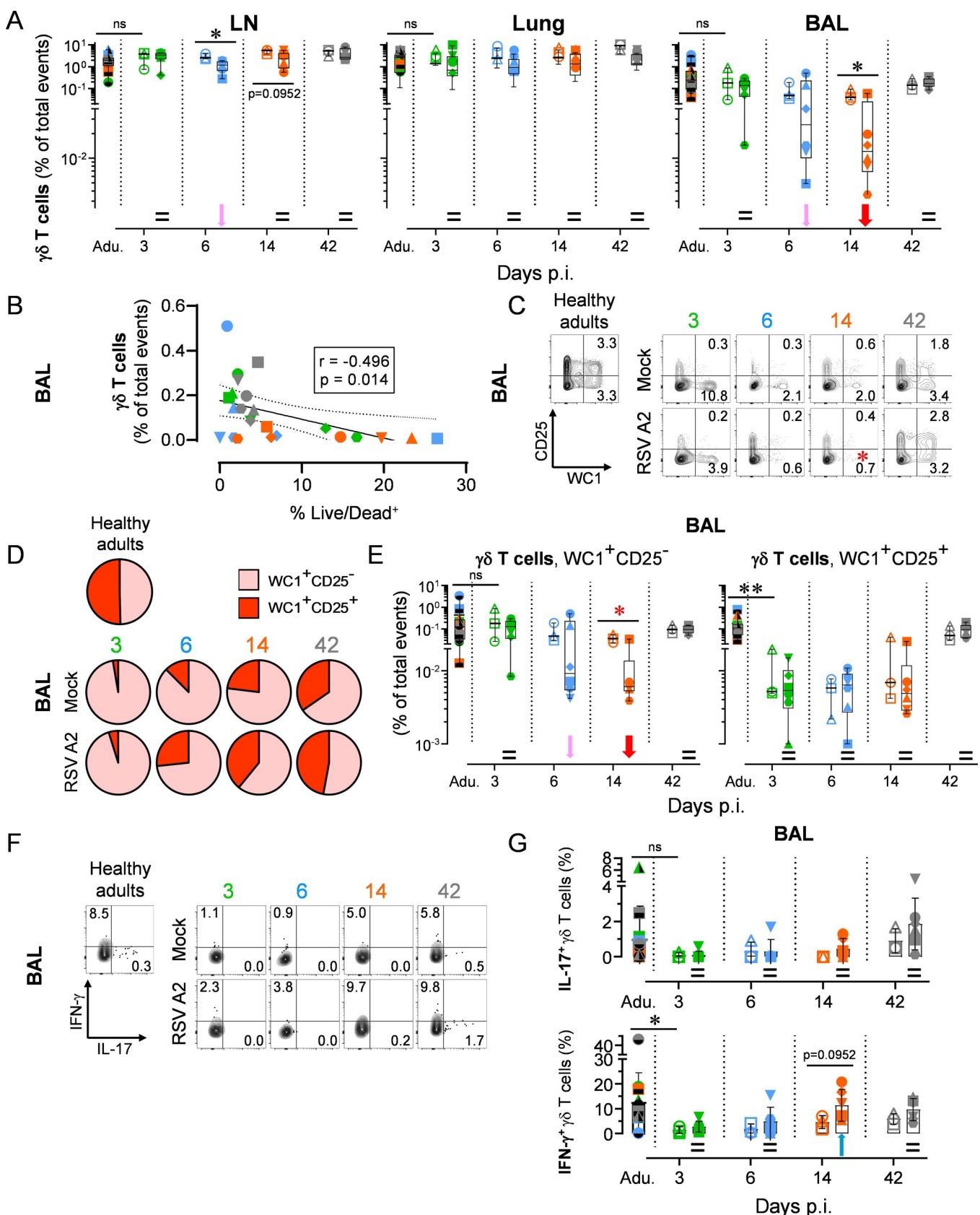

**Fig 3. Neonatal RSV A2 infection activates a transient depletion of γδ T cells and deficiency of IL-17-mediated immunity.** (A) Neonatal RSV A2 infection induces a decrease in γδ T cells number in the bronchoalveolar space, but not in the lung tissue. (B) Correlation coefficient (r) obtained with in γδ T cell counts calculated as a function of % Live/Dead$^+$. (C-E) Replenishment and maturation (CD25 up-regulation) of γδ T cell subset following the depletion after RSV A2 infection. WC1 is highly expressed in ruminant γδ T cells and is commonly used as a specific marker of this subset [68]. (C) Representative FACS contour plots. Numbers indicate the mean percentage of animals per group per time point. Red asterisk indicate significant differences between RSV A2 and Mock groups for a given time point. (D) As in (C) but with pie charts displaying all individuals. (E) Neonatal RSV A2 infection induces a decrease in γδ T cells number in the bronchoalveolar space. (F-G) Inability of γδ T cells to produce IL-17. (F) Representative FCM contour plots. Numbers indicate the mean percentage of animals per group per time point. (G) As in (F) but with plots displaying all individuals. (A-G) Each symbol represents an individual animal (healthy adults, n = 12; mock neonates, n = 3 per time point; neonates infected with RSV, n = 6 per time point). Boxplots indicate median value (center line) and interquartile ranges (box edges), with whiskers extending to the lowest and the highest values. Adu., healthy adults. Groups were compared using Mann–Whitney U tests (A, C, E-G). Stars indicate significance levels. $^*$, p < 0.05; $^{**}$, p < 0.01.

S6D and S6E). Surprisingly, mock neonates had high baseline frequencies of Th2 cells in BALs, which declined progressively over time. Elevated amounts of spontaneous produced IL-4 were a signature of the developing CD4$^+$ fraction, since it was no longer detectable at day 42, or in healthy adults. Upon infection, this analysis revealed an increase in the frequency of those Th2 cells, whereas there was no consistent influence on IFN-γ induction (Fig 4C (representative contour plots), 4D (interleaved scatters integrating all animals) and 4E (absolute counts)). Consequently, RSV infection in early life exacerbates the overall lung pro-Th2 cytokine environment, rather than inducing a Th2 immune response *per se*.

High baseline frequencies of IL-4-expressing CD8$^+$ T cells (Tc2) were also found in BALs of mock neonates. Again, a progressive decline of Tc2 cells occurred up to day 42 p.i., comparable to negligible levels measured in healthy adults. RSV disease induced a greater overall Tc2-derived cytokine environment in BALs. This exacerbation was tightly regulated, reduced to a level similar to healthy adults at day 14 p.i. Again, no consistent influence of RSV infection was seen on IFN-γ induction (Fig 4F (representative contour plots), 4G (interleaved scatters integrating all animals) and 4H (absolute counts)). When we compared the absolute numbers of recruited Th2 and Tc2 cells in BALs, we found that the IL-4 response at days 6–14 p.i. was mainly driven by the Tc2 cells (Fig 4I). Altogether, these results highlight the role of Tc2 cells as an important source of IL-4 production during neonatal RSV infection. Finally, we then tested the activation and differentiation of naïve T lymphocytes into memory subsets. The most striking observation was achieved at day 42 p.i.; the percent contributions of different subsets seen in RSV-infected neonates was highly superimposable to that found in healthy adults. This was not the case for mock neonates who had lower frequencies of activated memory T cells, particularly for the CD4$^+$ fraction (Fig 4J and 4K). Consequently, RSV infection in early life modulates the activation and the maturation of the T cell pool. Nonetheless, it is unknown whether this precipitated activation/maturation is RSV epitope-specific and a predictive factor for severe outcomes in the context of a secondary RSV infection [35].

## Neonatal RSV A2 infection rapidly induces Treg suppressive functions to dampen the Th2 and Tc2 responses

The neonatal immune system is tolerogenic, a mechanism that promotes self-tolerance to developing organs; fetal CD4$^+$ T cells are more likely to differentiate into Tregs after stimulation [10]. The shortened kinetics observed for CD4$^+$ T cells was elaborated by determining whether this was due to an early Treg rather than a classical CD4$^+$ T cell response (gating strategy in S7D Fig; % of total events in Fig 5A; absolute counts in S6F Fig). RSV infection elicited a rapid Treg recruitment in BALs that mirrored the one previously observed for CD4$^+$ T cells. This Treg dynamic is specific for the respiratory mucosa, since Treg frequencies remained unchanged in the lung tissue (Fig 5A). We then noticed a trend for Treg subset overrepresentation following RSV infection (Fig 5B). The proportion of Tregs declined progressively over

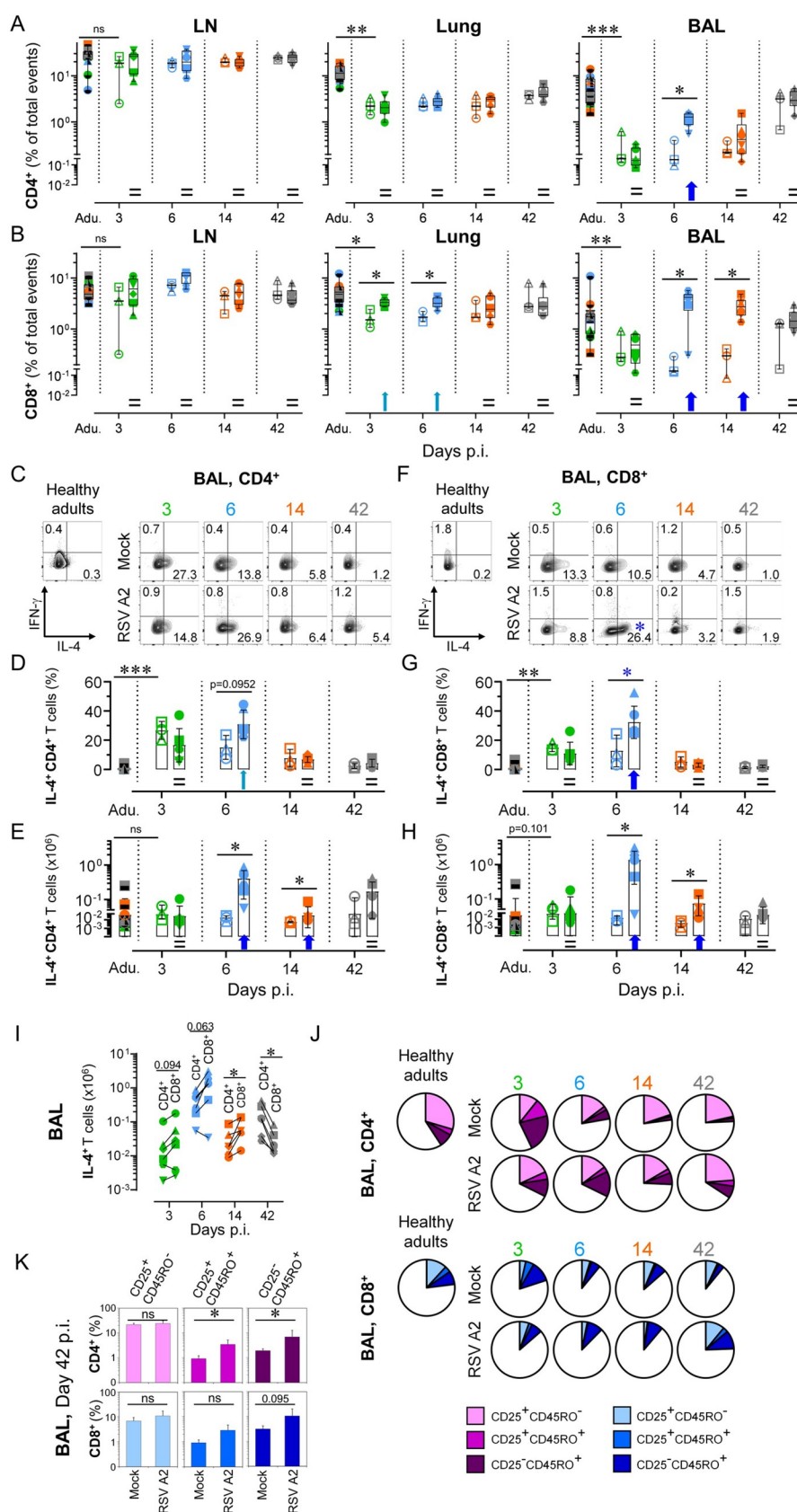

**Fig 4. Neonatal RSV A2 infection induces a fast and massive Tc2 influx in the bronchoalveolar space.** (A) RSV A2 infection in neonate accelerates the recruitment of CD4[+] in the bronchoalveolar space, but not in peribronchial LNs or lung tissue. (B) RSV A2 infection in neonate induces a strong recruitment of CD8[+] T cells in the bronchoalveolar space, and to a lesser extent in lung. (C-E) A high frequency of recruited CD4[+] T cells in bronchoalveolar space differentiate towards a Th2 cytokine profile (IL-4 production). (C) Representative FCM contour plots. Numbers indicate the mean percentage of animals per group per time point. (D) As in (C), but with plot displaying all individuals. (E) Absolute count of recruited IL4[+] CD4[+] T cells in bronchoalveolar space upon RSV A2 infection (F-H) A high frequency of recruited CD8[+] T cells in bronchoalveolar space differentiate towards a Tc2 cytokine profile (IL-4 production) (F) Representative FCM contour plots. Numbers indicate the mean percentage of animals per group per time point. Blue asterisk indicate significant difference between RSV A2 and Mock groups for a given time point. (G) As in (F), but with plot displaying all individuals. (H) Absolute count of recruited IL4[+] CD8[+] T cells in bronchoalveolar space upon RSV A2 infection. (I) Comparison of the absolute numbers of recruited Th2 (IL4[+] CD4[+]) and Tc2 (IL4[+] CD8[+]) cells in bronchoalveolar space upon RSV A2 infection. (J) RSV A2 infection in neonates modulates the activation (cell surface expression of CD25) and the maturation (cell surface expression of T cell memory marker CD45RO) of CD4[+] and CD8[+] T cell pools in the lung. (K) As in (J), but with histograms integrating all individuals. (A-K) Each symbol represents an individual animal (healthy adults, n = 12; mock neonates, n = 3 per time point; neonates infected with RSV, n = 6 per time point). Boxplots indicate median value (center line) and interquartile ranges (box edges), with whiskers extending to the lowest and the highest values. Adu., healthy adults. Groups were compared using Mann–Whitney U (A-H, K) and Wilcoxon tests (I). Stars indicate significance levels. *, $p < 0.05$; **, $p < 0.01$; ***, $p < 0.001$.

time, tending towards similar levels to those detected in healthy adults. RSV infection promoted enhancement of TGF-β production by Tregs (Fig 5C (representative contour plots), 5D (interleaved scatters integrating all animals) and 5E (absolute counts)). When Treg counts were calculated as a function of non-Treg CD4[+] counts, an association was found, suggesting that CD4[+] T cell pool tends to differentiate into Tregs. When Treg counts were calculated as a function of CD8[+] counts, an association was still found, confirming that CD8[+] T cells and Tregs colonized the alveolar space in a coordinated manner. By plotting CD8[+] count as a function of non-Treg CD4[+] counts, it appeared that T cell response was mainly driven by the CD8[+] fraction (high numbers at days 6 and 14 p.i.) (Fig 5F).

## Neonates and adults are mounting a distinct pulmonary T cell response following RSV A2 infection

We followed the same approach to dissect the cellular immune response in RSV A2-infected adults. Time-points were selected as follow: 6 days p.i. to evaluate the innate responses; 14 days p.i. to assess the adaptive mechanisms and virus clearance (Fig 6A). Additionally, neonates were infected with RSV-ON1-H1 strain, a primary clinical isolate recently circulating in central Europe (see following section) [36]. BAL cellularity is provided in S8 Fig; no significant difference was found between the absolute counts of RSV-infected adults and neonates. While the viral loads and virus clearance was comparable between adults and neonates (Fig 6B), respiratory lesions were absent in adults (Fig 6C); this suggests that the increased rate of severe disease reported in infants is not due to a failure to control the virus, reinforcing the hypothesis for immune-mediated RSV pathogenesis. Interestingly, the induction of an efficient RSV-neutralization was found in the serum of both RSV A2-infected adults and neonates (Fig 6D).

The rapid and robust pDC recruitment observed in infected neonates did not occur in infected adults. However, lung-resident pDCs of the latter were prompt to react, displaying an even more mature and activated phenotype in 2 out of 3 animals (Fig 7A, 7B and 7C). The strong decrease in γδ T cell counts observed in infected neonates was absent in adults. Moreover, adult γδ T cells produced high amounts of IL-17, suggesting that IL-17-mediated immunity is deficient in early life (days 3–14 p.i.; Fig 7D, 7E and 7F) [15]. This IL-17 response was accompanied by an activated phenotype, as indicated by the enhanced proportion of CD25[+] cells (Figs 7G (pie charts displaying all individuals) and S9A (interleaved scatters integrating all animals)). From these results, we could ascertain that neonates lack a mature and functional

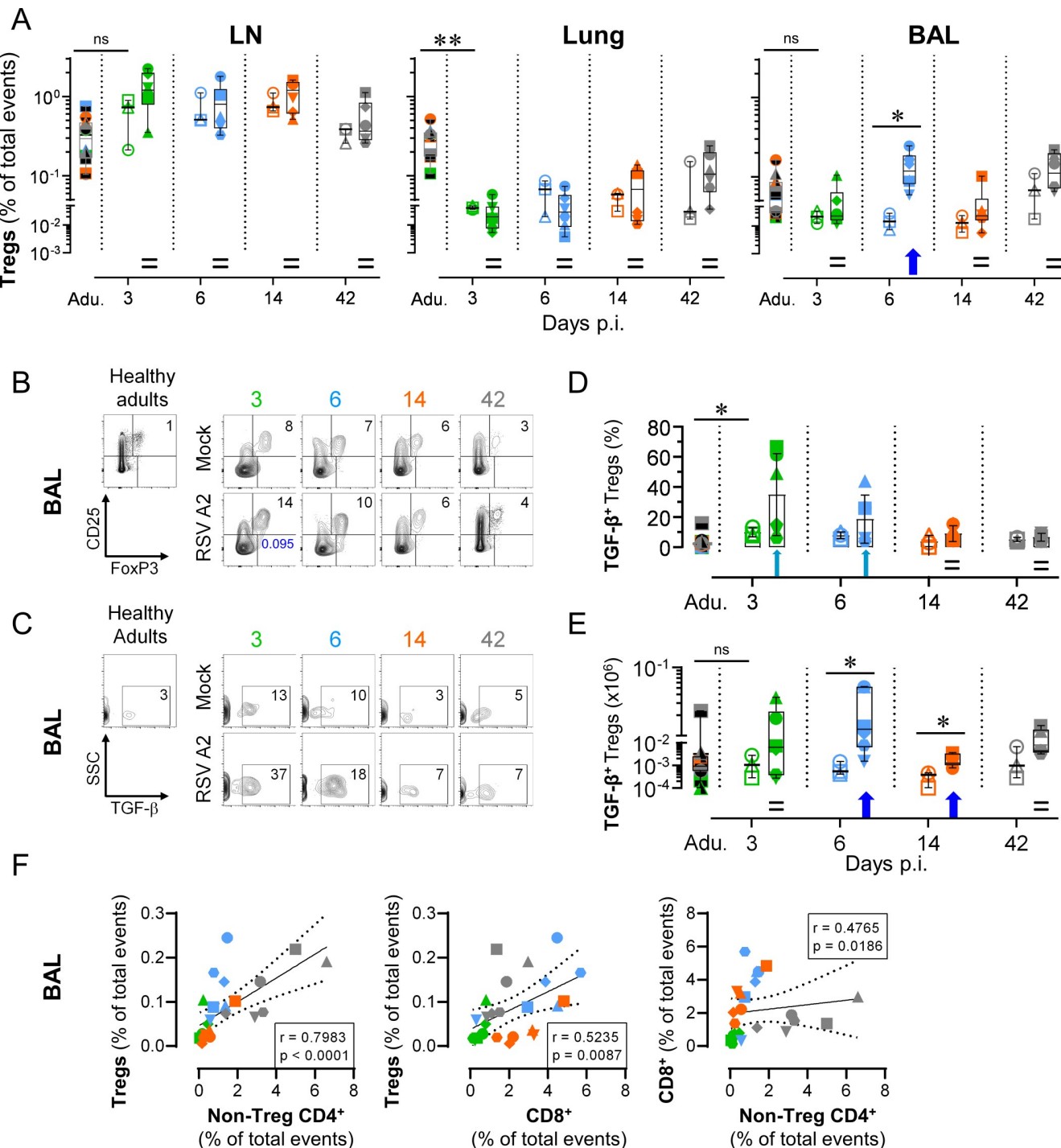

**Fig 5. Neonatal RSV A2 infection rapidly induces Treg suppressive functions to dampen the Th2 and Tc2 responses.** (A) RSV A2 infection in neonates led to the recruitment of Tregs in the bronchoalveolar space, but not in peribronchial LNs or lung tissue. (B) Progressive decrease with age of Treg subset among the CD4[+] T cell fraction (representative FCM contour plots). Numbers indicate the mean percentage of animals per group per time point. Number in blue indicate a trends to increase at day 3 p.i. (C) Representative contour plots with the frequency of TGF-β-producing Tregs. Numbers indicate the mean percentage of animals per group per time point. (D) As in (C), but with plot displaying all individuals. (E) Absolute count of recruited TGF-β[+] Tregs in bronchoalveolar space upon RSV A2 infection. (F) Correlation coefficient (r) obtained in BALs with Treg counts calculated as a function of non-Treg CD4[+] counts (left), Treg counts calculated as a function of CD8[+] counts (middle) and CD8[+] counts calculated as a function of non-Treg CD4[+] counts (right). Significance was reached for the three correlations. (A-F) Each symbol represents an individual animal (healthy adults, n = 12; mock neonates, n = 3 per time point; neonates infected with RSV, n = 6 per time point). Boxplots indicate median value (center line) and interquartile ranges (box edges), with whiskers extending to the lowest and the highest values. Adu., healthy adults. Groups were compared using Mann–Whitney U test (A-E). Stars indicate significance levels. *, p < 0.05; **, p < 0.01.

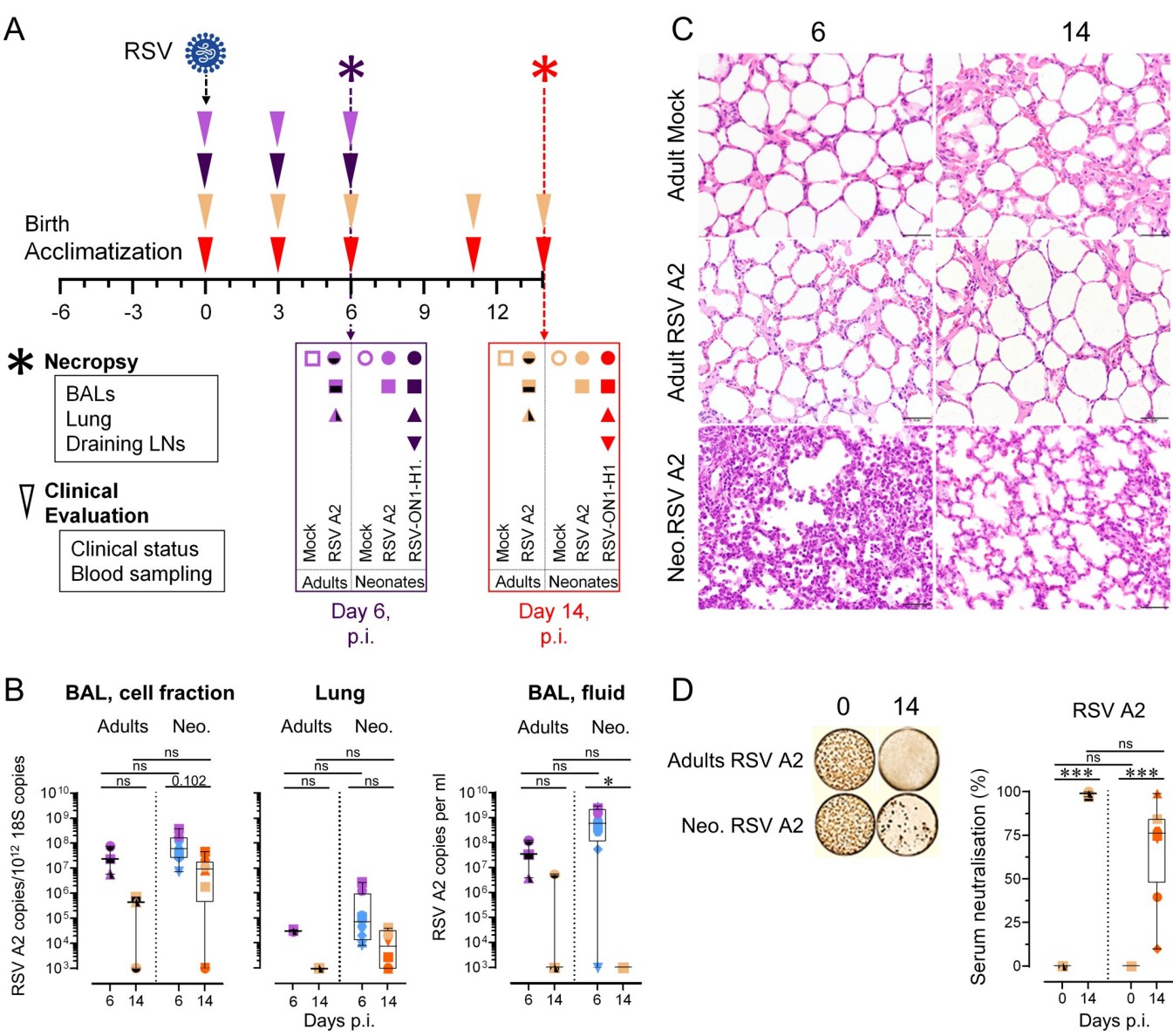

**Fig 6. RSV A2-infected adults clear efficiently the virus in absence of lung lesions.** (A) Experimental design of the trans-tracheal RSV A2 or RSV-ON1-H1 infections. Animals were injected with $10^8$ PFU (RSV A2) or $10^7$ PFU (RSV-ON1-H1). As controls, PBS-injected newborns or non-infected adults were used. Before and after RSV infection, clinical evaluation was assessed by a veterinarian, every 3 days. Necropsies were done at days 6 or 14. (B) RSV A2 was quantified in the respiratory tract (BAL, cell fraction; lung; BAL, fluid) of adults and neonates by qPCR. (C) Representative histological lung tissue sections from RSV A2-infected neonates and adults at day 6 and 14 p.i. (40X, HE). Adults show no histological lesions, whereas neonates show interstitial pneumonia at day 6.p.i. (D) Adults and neonates produced RSV A2 NAbs following infection. Serums collected in animals prior RSV A2 infection (0), and at days 14 p.i. (14), were co-incubated with RSV A2 (100 PFU) and applied to HEp-2 cells for 48 hours. (A-D) Each symbol represents an individual animal (RSV A2 infected adults, n = 3; Neonates infected with RSV A2, n = 8 per time point). Boxplots indicate median value (center line) and interquartile ranges (box edges), with whiskers extending to the lowest and the highest values. Groups were compared using one-way ANOVA followed by Turkey's post hoc test (B, D). Stars indicate significance levels. *, p < 0.05; ***, p < 0.001.

γδ T cell pool in the lower airways. For the CD4+ subset, no significant recruitment was found when compared to related healthy adults (Fig 7H). The percentages of IL-4-producing cells were far from achieving the magnitude observed in neonates, and RSV infection had no consistent influence on IFN-γ induction (Fig 7I and 7J). Another clear difference was the contraction of effector memory pool (CD25+CD45RO+) at day 14 p.i., indicative of a specific immune

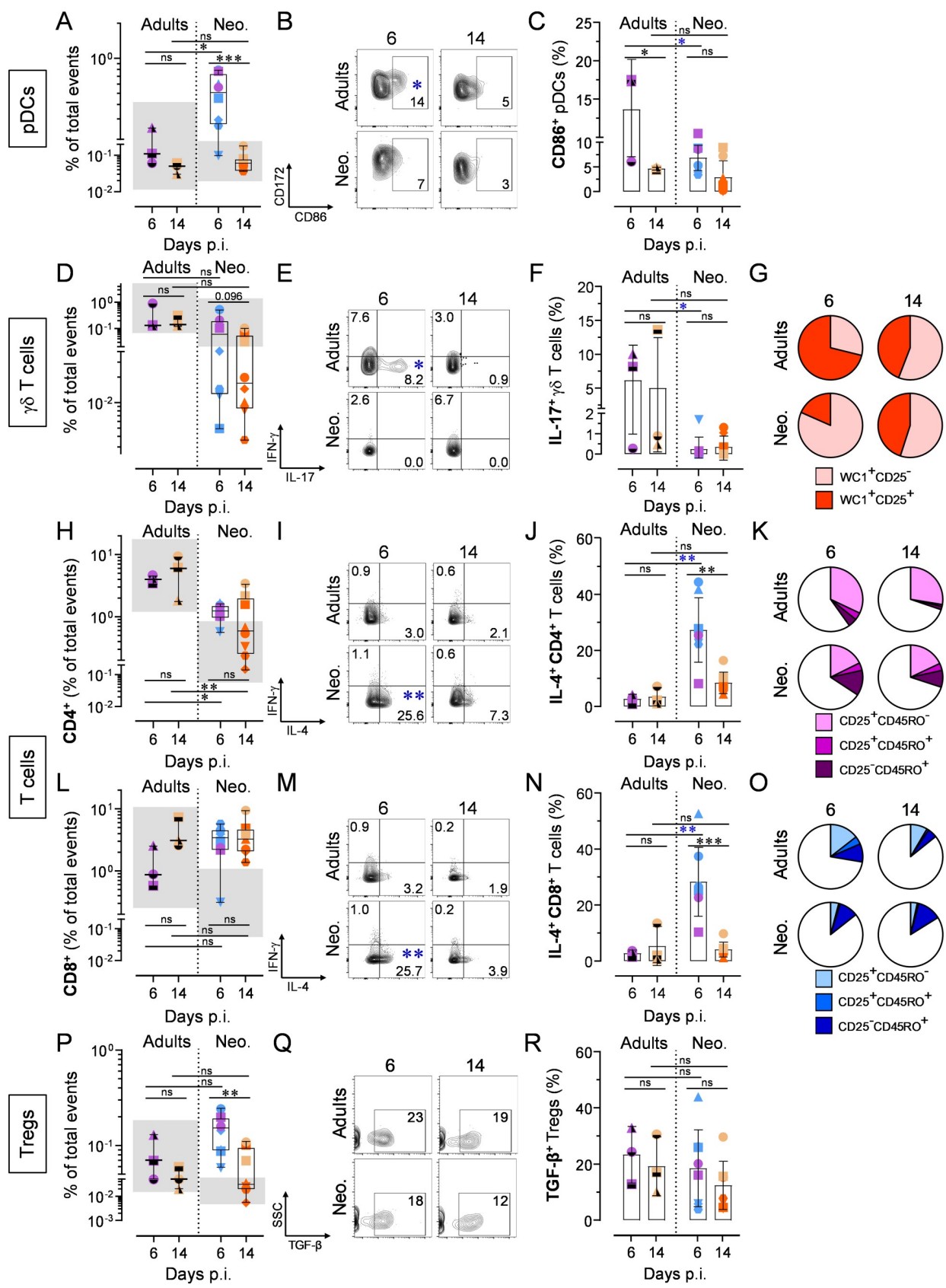

**Fig 7. Distinct cellular immune response in neonates compare to adults following RSV A2 infection.** (A-C) Counts of recruited pDCs and CD86 upregulation in bronchoalveolar space. (A) Counts of recruited pDCs in bronchoalveolar space upon RSV A2 infection. (B) Representative FCM contour plots showing CD86 upregulation. Numbers indicate the mean percentage of animals per group per time point. (C) As in (B), but interleaved scatters integrating all animals. (D-G) γδ T cell counts, cytokine production, and maturation in the bronchoalveolar space. (D) Counts of γδ T cells in bronchoalveolar space upon RSV A2 infection. (E) Representative FCM contour plots showing IL-17 secretion. Numbers indicate the mean percentage of animals per group per time point. Blue asterisk indicate significant difference between RSV A2-infected adult and neonate groups for a given time point. (F) As in (E), but interleaved scatters integrating all animals. (G) Pie charts integrating all animals. (H-K) Recruitment, activation, and maturation of the CD4$^+$ T cell pool in the bronchoalveolar space. (H) Counts of CD4$^+$ T cells in bronchoalveolar space upon RSV A2 infection. (I) Representative FCM contour plots showing IL-4 secretion. Numbers indicate the mean percentage of animals per group per time point. Blue asterisk indicate significant difference between RSV A2-infected adult and neonate groups for a given time point. (J) As in (I), but interleaved scatters integrating all animals. (K) Pie charts integrating all animals. (L-O) As in (H-K), but showing CD8$^+$ T cell pool. (P-R) Recruitment of Tregs and TGF-β production in the bronchoalveolar space. (P) Counts of recruited Tregs in bronchoalveolar space. (Q) Representative FCM contour plots showing TGF-β secretion. Numbers indicate the mean percentage of animals per group per time point. (R) As in (Q), but interleaved scatters integrating all animals. (A-R) Each symbol represents an individual animal (infected adults, n = 3 per time point; neonates infected with RSV, n = 8 per time point). The grey zone corresponds to the interval integrating the values of all corresponding mock controls. Boxplots indicate median value (center line) and interquartile ranges (box edges), with whiskers extending to the lowest and the highest values. Groups were compared using one-way ANOVA followed by Turkey's post hoc test (A, C, D, F, H, J, L, N, P, R). Stars indicate significance levels. *, p < 0.05; **, p < 0.01; ***, p < 0.001.

response (Figs 7K and S9B). The recruitment of CD8$^+$ T cells was delayed compared to neonates, with just a trend only detectable at day 14 p.i. (Fig 7L). Again, the percentage of IL-4-producing cells was low compared to RSV A2-infected neonates (Fig 7M and 7N), whereas the precipitous contraction of effector memory pool at day 14 p.i. suggested a RSV-specific CD8$^+$ T cell response (Figs 7O and S9B). From these results, it is likely that the inaptitude of neonates to mount a classical effector T cell response is tentatively substituted by a Th2/Tc2 cell recruitment. To elucidate this point, the follow-up of the present work will consist in evaluating the RSV-specificity of this neonatal type 2 responses. Finally, we evaluated the Treg compartment in adults. Unlike our previous observations in neonates, we failed to detect any Treg recruitment in the bronchoalveolar space of infected adults, at least up to day 14 p.i. However, the rare tissue-resident Tregs were able to produce TGF-β, suggesting that an immune regulation happened (Fig 7P, 7Q and 7R).

## The severity of RSV disease relates to the magnitude of the bronchoalveolar T cell responses

While immune-mediated mechanisms are central determinants of RSV severity, there are indications that viral factors play also a role [37]. We evaluated the pulmonary T cell response in neonates infected with RSV A2, a well characterized prototypical strain of RSV, in comparison to the RSV-ON1-H1, a primary clinical isolate which was the major genotype circulating in central Europe during the RSV seasons of 2012 to 2017 (Fig 6A) [36]. Whereas infection of neonates with RSV-ON1-H1 strain led to a milder disease, as indicated by milder histological lesions (Fig 8A), the viral loads and virus clearance was comparable between both strains, again reinforcing the hypothesis for immune-mediated RSV pathogenesis (Fig 8B). Effectively, all T cell-related responses were either attenuated or displayed a trend towards attenuation. We did not observe a γδ T cell depletion, and the CD8$^+$ T cell recruitment was delayed (requiring 14 days p.i. to exceed the limits of the Mock grey zone, instead of 6 days p.i. in RSV A2-infected neonates). Furthermore, the Treg recruitment was significantly lower compared to RSV A2 infection (Fig 8C). By combining results obtained with both strains, we found a significant association between Treg and pDC counts in BALs (Fig 8D). This indicates that, despite their skew, all distinct subsets of neonatal T cells cooperate tightly with the magnitude of the cellular immune response against RSV infection. Altogether, our data clarified how neonates elicit an over-exuberant pulmonary T cell response characterized by a bronchoalveolar influx of Treg, Th2, and Tc2 cells, contrasting to a classical T cell response.

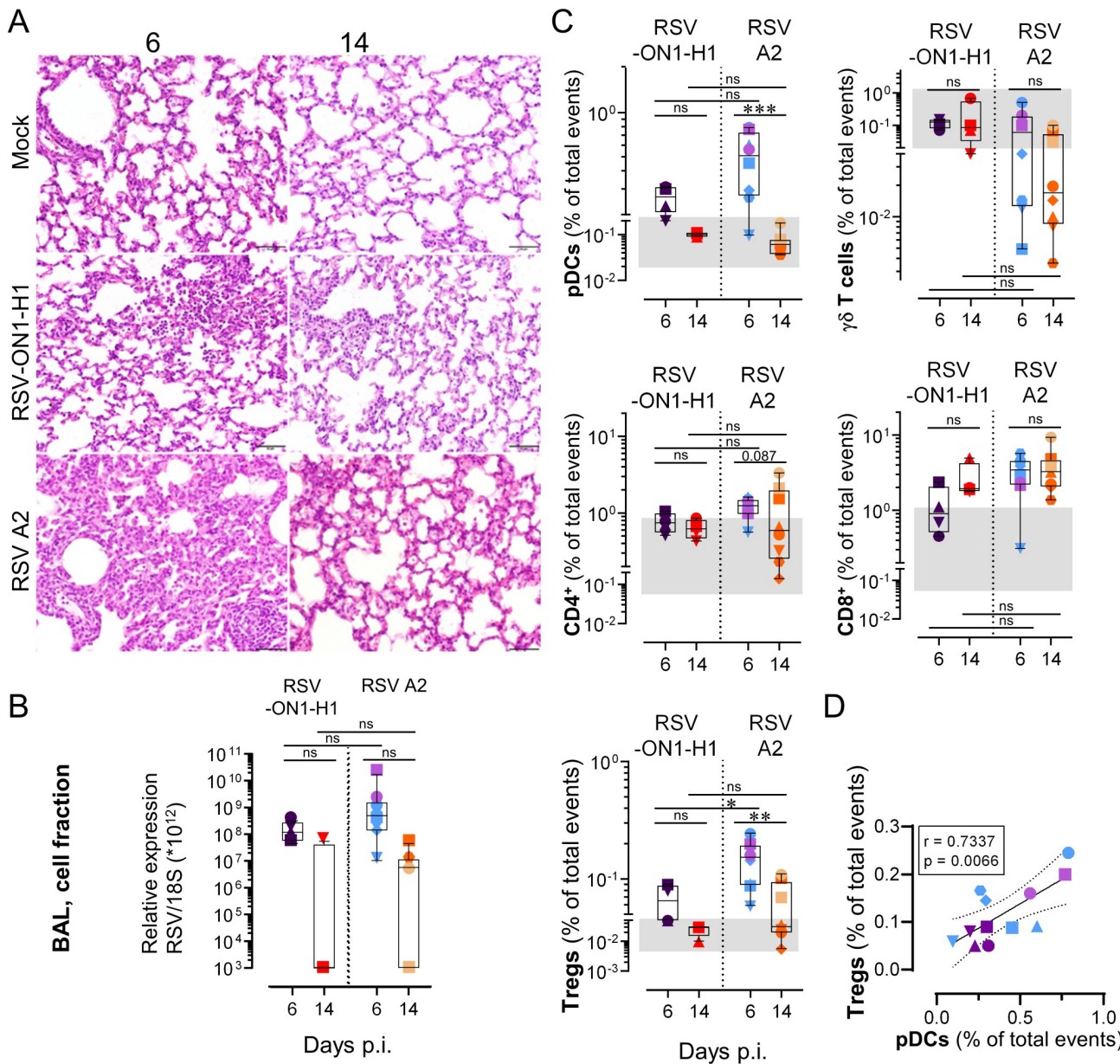

**Fig 8. The severity of disease relates to the magnitude of the unconventional bronchoalveolar T cell responses.** (A) Representative histological lung tissue sections from mock neonates and neonates infected with RSV-ON1-H1 or RSV A2 strain (40X, HE). At day 6 p.i. the RSV-ON1-H1 infected animals show mild interstitial pneumonia (upper right corner) with accumulation of inflammatory cells within alveoli. No lesions at day 14 p.i. or mock-infected animals. (B) RSV was quantified in the respiratory tract (BAL, cell fraction) of neonates infected with either RSV-ON1-H1 or RSV A2 strain by qPCR. (C) Comparison of immune cell subset trafficking into bronchoalveolar space after infection with RSV-ON1-H1 or RSV A2 strains. (D) Correlation coefficient (r) obtained in BALs with Treg counts calculated as a function of pDC counts. (B-D) Each symbol represents an individual animal (neonates infected with RSV-ON1-H1, n = 4 per time point; neonates infected with RSV A2, n = 8 per time point). The grey zone corresponds to the interval integrating the values of all corresponding mock controls. Boxplots indicate median value (center line) and interquartile ranges (box edges), with whiskers extending to the lowest and the highest values. Groups were compared using one-way ANOVA followed by Turkey's post hoc test (B, C). Stars indicate significance levels. *, p < 0.05; **, p < 0.01; ***, p < 0.001.

## Discussion

To properly assess the immunopathogenesis of RSV, it is essential to use a model that recapitulates the features of human disease. We selected the neonatal lamb, as RSV is a natural pathogen for this species, lung development is similar, and mimic the key features of human RSV infection

in infants [27,29]. One strength of our experimental approach was the ease sampling and the full accessibility to the lower airways throughout the disease when sampling in infants is mainly restricted to the upper airways, the latter representing only an approximate surrogate of the distal lung where RSV disease occurs. Moreover, we offer a view of the development of pulmonary neonatal immunity in healthy conditions. This process is highly dynamic and tightly regulated, with a short and early stage where colonizing T cell subsets synergize towards a narrow pro-tolerogenic window, namely a "Treg/Th2/Tc2" environment. This certainly contributes to the maturation of the immune system and may constitute an unsuitable state to resolve RSV infection.

The mechanisms underlying the severity of disease are still unclear, although RSV itself was shown to induce distal lung obstruction by shedding of necrotic airway epithelial cells [38]. Also, it is unclear yet if RSV disease severity is associated with viral loads [39,40]. Here we demonstrate that while RSV disease was more severe in neonates compare to adult animals, RSV loads and virus clearance was comparable for both age groups. In line with this, RSV loads and virus clearance were similar for neonates infected with RSV A2 or RSV-ON1-H1 strains inducing a disease with distinct severity. Clearly, our data reinforce the hypothesis of an inappropriate neonatal immune response as a determinant of RSV disease severity. We observed a rapid and robust recruitment of neonatal pDCs, whose extent correlated with the viral loads of infected neonates. Infected adults did not require this recruitment, since their lung-resident pDCs reacted promptly by maturing to a higher level than that measured in neonates. This partially corroborates recent studies reporting low counts and maturation levels of pDCs in mice and infants [41,42]. This reduced capacity to respond against RSV might be due to a direct disturbance of signaling by RSV itself [43], or an intrinsic deficiency of neonatal pDCs to produce elevated amounts of cytokines [44–46]. This neonatal defect would prevent conventional DCs from adequate presentation of viral antigens, contributing to poor specific T cell responses [47].

Controversial is whether γδ T cells participate in viral clearance or immunopathology. Herein, the γδ T cell depletion conflicted with a previous report. Indeed, children with severe RSV bronchiolitis had reduced frequencies of γδ T cells in peripheral blood, which was explained by the authors as a likely redistribution towards the lungs; unfortunately, they could not evaluate their hypothesis [48]. In line with our findings, mice depleted of γδ T cells before RSV infection display increased viral titers [13]. An alternative hypothesis that would reconcile our study with others, is that γδ T cell tissue redistribution plays less of a role than its immature state in early life. Peripheral blood γδ T cells of infants with severe disease failed to produce IFN-γ when restimulated [14], whereas γδ T cells from neonatal mice had an impaired ability to produce IL-17 [15]. Finally, Tc17 and Th17 cells were recently associated with shorter hospital stays and proposed to play a protective role [26]. Altogether, our data demonstrate that the protective role of γδ T cells in RSV infection is largely inefficient in early life.

The BALs of healthy neonates displayed a high proportion of Tregs that declined rapidly to levels comparable to healthy adults. As fetal CD4$^+$ T cells tend more to differentiate into Tregs after stimulation than in adults [10], this tolerogenic mechanism is likely to promote self-tolerance to the developing lung. We found that Treg response to RSV infection happened very early in neonates, accompanying CD4$^+$ and CD8$^+$ T cell expansion rather than shutting them down afterwards. This strong Treg recruitment was absent in adults. We speculate that the neonate immune system faces a dilemma between avoiding potential autoimmune disorders and failing to mount a specific T cell response to harmful pathogens. Studies conducted in young infants with RSV disease support this statement. A selective depletion of peripheral Tregs was shown, probably due to massive recruitment to the lungs [49]. A higher level of TGF-β transcript was measured in neonatal DCs compared to adults [50]. In contrast, some reports emphasized the protective role of Tregs, since Treg-depleted young mice had an enhanced RSV disease, with abundant T cells exhibiting an activated phenotype [51]. Another

remarkable observation was the strong correlation between pDC and Treg counts. Despite the need for further investigations, it is worth noting that pDC to Treg ratio was linked to the clearance *versus* persistence of human papillomavirus [52].

Whether the cellular arm of the adaptive pulmonary response is detrimental or beneficial in RSV disease is still debated. Particularly, the contribution of IL-4 to viral clearance versus immunopathogenesis remains controversial [25,53–55]. Recently, a study performed in hospitalized infants showed a possible involvement of Tc2 cells in disease severity. Although convincing, this study was for obvious ethical reasons restricted to nasal aspirates [26]. Interestingly, Tc2 cells are increasingly proposed to play an important role in the etiology of asthma [56]. Herein, we provide a significant advancement in understanding this yet unclear IL-4 contribution, by showing that type 2 cytokine environment in the lung is a footprint of early life that decreases rapidly with time in healthy conditions. RSV infection circumvents this decline by mediating the massive recruitment of CD4$^+$ and CD8$^+$ T cells, exacerbating the overall type 2 cytokine environment. Also, one must consider the key role of type 2 innate lymphoid cells (ILC2s), a scarce innate subset with lymphoid morphology but distinct from T cells, in promoting IL-4 secretion in the BALs of RSV-infected neonates [55,57,58]. This direct investigation was prevented by the lack of ovine antibodies to constitute the accurate FCM panel for a proper investigation. However, a proportion of IL-4-producing CD4$^-$CD8$^-$ cells was detected, supporting the notion that ILC2s may participate to the early induction of type 2 cytokines. Based on our results, we think that the link between IL-4 and RSV disease is the direct consequence of a trafficking of endogenous Th2 and Tc2 lymphocytes towards infected pulmonary tissues, rather than an RSV-specific type 2 immune response *per se*.

The high proportion of pulmonary T cells producing spontaneously IL-4 was undetectable in the healthy adults. Neither was the RSV-specific immune response of adults oriented towards a Th2 or Tc2 profile. This corroborates a previous report mentioning an endogenously poised cytokine profile toward IL-4 in CD4$^+$ T cells in cord blood of neonates and infants, establishing a link between IL-4 and development [59]. However, we provide new insights by demonstrating that i) CD8$^+$ Tc2 cells are a superior source of IL-4 than CD4$^+$ Th2 cells; ii) this immunological state occurs during a short period *in vivo*, raising the likelihood that numerous animal trials failed to detect it due to the study design. Remarkably, this transient type 2 cytokine environment was exacerbated upon RSV infection. So far, similar observations were only made at the systemic level or in nasal aspirates, and authors could only speculate on the relocation to the lower respiratory tract [26,60]. Herein, not only we show that this happens, but we also unify the link between IL-4 and RSV disease and endogenous IL-4 program in neonates. This finding constitutes an important step towards deciphering why neonates and young infants are particularly at risk.

In conclusion, we provide novel insights into the identification of the host's contributing factors to RSV immunopathology, strengthening the hypothesis of an inappropriate immune response in the narrow temporal window of early life. This can now be characterized as a combined bronchoalveolar influx of Treg, Th2 and Tc2 cells, associated with a strong depletion of γδ T cells and IL-17 deficiency. These findings constitute a novel basis for further exploration of RSV immunopathogenesis in infants and should be considered in RSV vaccine design, which remains challenging after five decades of effort.

## Material and methods

### Ethics statement

The experiments in lambs were performed in compliance with the Swiss animal protection law (TSchG SR 455; TSchV SR 455.1; TVV SR 455.163) under the authorization BE125/17. The

experiments were reviewed by the cantonal committee on animal experiments of the canton of Bern, Switzerland, and approved by the cantonal veterinary authority (Amt für Landwirtschaft und Natur LANAT, Veterinärdienst VeD, Bern, Switzerland). For the RSV-ON1-H1 strain isolation, the caregivers gave written informed consent for the donation of nasopharyngeal aspirates, and all steps were conducted in compliance with good clinical and ethical practice and approved by the local ethical committee at Hannover Medical School, Germany (permission number 63090/2012)

## Virus

Human RSV A2 strain was obtained from the American Type Culture Collection (ATCC, VR-1540, GenBank accession number KT992094.1). The strain RSV-ON1-H1 was isolated from nasopharyngeal aspirates of a child below the age of five years with confirmed RSV infection, hospitalized at Hannover Medical School, Germany [61]. Both RSV strains were propagated on HEp-2 cells (ATCC, CCL-23) cultured in DMEM (Gibco) supplemented with 5% FBS (Gibco), at 37˚C in 5% CO2 atmosphere. Briefly, viruses were incubated for 36–44 h in tissue culture flasks at a multiplicity of infection (MOI) of 0.02 plaque-forming unit (PFU) per cell. When the cytopathogenic effect (CPE) reached approximately 60%, the cultures were frozen overnight at −70˚C, to disrupt the HEp-2 cells and release the virus. Then, the culture flasks were thawed, HEp-2 cells were scraped and the complete contents of the flask were spun down at 1000 g, 10 min, 4˚C to collect cell debris. The clarified supernatants were layered over 20% sucrose (w/v) solution prepared in sterile PBS and ultra-centrifuged for 3 h 30 min at 71'000 g at 4˚C. The resultant RSV stocks were aliquoted and stored at −150˚C until further use. Virus titers were determined on HEp-2 cells, using biotinylated anti-RSV antibody (goat polyclonal IgG, Bio-Rad) for 1 h at RT. The plaques were visualized using ExtrAvidin-Peroxidase (Sigma), 30 min at RT, and revealed by DAB substrate (Sigma), left for up to 30 min at RT. When the plaques appeared with the desired coloration, the reaction was stopped with PBS. RSV titers were expressed as PFU per ml.

## Primary ovine well-differentiated airway epithelial cell cultures and RSV A2 infection

Ovine well-differentiated airway epithelial cell (WD-AECs) at the air-liquid-interface (ALI) were established as described with some modifications [62]. Shortly, AECs were isolated by enzymatic digestion of ovine tracheobronchial tissue with 1 mg/ml protease XIV from *Streptomyces griseus* (Sigma) and 10 μg/ml DNase I (Worthington). AECs were expanded in culture using expansion medium 1 formulated as follow: PneumaCult-*Ex Plus* Medium (StemCell Technologies) supplemented with 100 μg/ml Primocin (Invivogen), 1 μM hydrocortisone (StemCell Technologies), and 10 μM Y-27632 (StemCell Technologies). Porous inserts of 0.4 μm (Greiner) were coated overnight with 100 μg/ml bovine type I collagen (Advanced Biomatrix) and 100'000 cells per insert were seeded in expansion medium 1. Inserts were kept undisturbed for 2–3 days at 37˚C, 5% CO$_2$, afterwards, medium was changed in the apical and basolateral chambers to expansion medium 2 formulated as follow: PneumaCult-*Ex Plus* Medium supplemented with 100 μg/ml Primocin, 1μM hydrocortisone, 10 μM Y-27632, 1 μM A 83–01 (Tocris), and 3 uM isoproterenol (Abcam). When confluence reached, the apical medium was removed and ALI medium composed of PneumaCult-*ALI* Medium (StemCell Technologies) supplemented with 4 μg/ml heparin (StemCell Technologies), 1 μM hydrocortisone, 100ug/mL Primocin, 25 ng/ml hEGF (Repligen) and 1 μM DAPT (StemCell Technologies) was added to the basolateral chamber to induce differentiation of the cells. Next, the medium was changed every 2–3 days until the appearance of ciliated cells and mucus. The cell

layer was washed twice a week with Hank's Balanced Salt Solution (HBSS, Gibco) to get rid of excessive mucus. WD-AEC cultures were infected apically with RSV A2 at a MOI of 1 PFU per cell, by applying the virus to the apical chamber for three hours at 37˚C, 5% $CO_2$), followed by washing three times with PBS (Gibco). Finally, at 24, 48, 72 and 96 hours p.i. the apical washes were harvested and stored at -80˚C until RSV A2 titration was performed.

## Animals

For the duration of the experiment, the animals were housed in a high containment facility and the *postpartum* acclimatization lasted 3 to 6 days before RSV infection. In accordance with the principles of the 3Rs, suckling lambs were held together with their adult mothers and fed *ad libitum*. The animals were kept on straw with a local ambient temperature of 20–22˚C. In order to prevent secondary bacterial infection, all the animals received a prophylactic long-acting oxytetracycline treatment starting at 18 to 24 h before RSV infection by intramuscular injection of Cyclosol LA, 20 mg/kg (Dr. E. Graeub AG), repeated after 4 days. The trans-tracheal RSV inoculation was performed under sedation and analgesia by intramuscular injection of a mixture of midazolam (Dormicum, Roche) 0.2 mg/kg and butorphanol (Morphasol, Dr. E. Graeub AG) 0.2 mg/kg. The injection site was shaved and disinfected with 70% ethanol. The trachea was punctured between the proximal tracheal rings with a 0.9 x 40 mm needle (20G, BD) and a volume of 2 ml of virus (RSV-infected animals: $10^8$ PFU for RSV A2; $10^7$ PFU for RSV-ON1-H1) or PBS (mock-treated controls) were injected slowly. Before and after infection, we monitored body temperature, and clinical status daily. The clinical status of the lambs was assessed by a veterinarian, whenever possible always the same person to ensure unbiased clinical assessment. The rectal body temperature was measured with a digital thermometer, respiratory and heart rates with a stethoscope and nasal swabs (4N6FLOQSwabs, Thermo-Fisher) were collected daily.

## Histopathology and Immunohistochemistry

Samples of each lung lobe were fixed in 10% buffered formalin for 48 h, embedded in paraffin and routinely processed for histology. Three μm histological sections were stained with hematoxylin and eosin. For immunohistochemistry, unstained 3 μm sections were deparaffinized with xylol for 5 minutes followed by rehydration in descending concentrations of ethanol (100, 95, 80, and 75%). Endogenous peroxidase activity was inhibited by $H_2O_2$ (3.25% in methanol, 10 min at RT). Then, heat-induced antigen retrieval was carried out by incubating the slides in boiling citrate buffer (pH 6.0) for 10 min. Nonspecific binding of antibody was blocked with 1% BSA for 10 min. Slides were incubated overnight at 4˚ with primary antibodies diluted 1:200 in PBS (clone 302, ThermoFisher). LSAB and AEC Kits (DakoCytomation) were used as indicated by the manufacturer for secondary antibody incubation and signal detection. Slides were counterstained with Ehrlich hematoxylin and cover slips were mounted using Aquatex (Merck) [63].

## Cell suspension preparation

The isolation of cells from peribronchial LNs was done using a Collagenase D (Sigma) and DNAse I (BioConcept) enzyme mix and the gentleMACS Octo Dissociator (Miltenyi Biotec). For the isolation of immune cells from the lung tissue, we used a Collagenase I (BioConcept), Collagenase II (BioConcept) and DNase I (BioConcept) enzyme mix. The dissociated tissues were passed through a 100 μm cell strainer (Falcon) and centrifuged at 250 g for 10 min, 4˚C. Next, the cell pellets were resuspended, passed through a 70 μm cell strainer (Falcon) to ensure the removal of clumps, and centrifuged 250 g for 10 min, 4˚C. For the lung suspensions, the

cells were further purified using the Ficoll-Paque approach (Sigma). For the isolation of cells from bronchoalveolar lavages (BALs), the lungs were taken out with the trachea, which was clamped before cutting, to prevent blood from entering the lungs. Then, PBS was poured through the trachea using a sterile funnel. The single-cell suspensions were applied to a sieve to remove any remaining larger particles. Depending on the BAL volume, single-cell suspensions were distributed in the appropriate number of 50 ml tubes for centrifugation at 350 g for 10 min, 4˚C. Then, the pellets were resuspended, passed through a 100 µm cell strainer (Falcon) and centrifuged at 350 g for 10 min, 4˚C. Next, the pellets were resuspended, passed through a 70 µm cell strainer (Falcon) to ensure the removal of clumps, and centrifuged at 350 g for 10 min, 4˚C. If needed, red blood cells were lysed by resuspending the pellet with 2 ml of $H_2O$ and washed immediately in cold PBS/EDTA, before centrifugation at 350 g for 10 min, at 4˚C.

## Human RSV neutralization assay

Serial 2-fold dilutions of heat-inactivated sera (30 min, 56˚C) were prepared in serum-free DMEM medium, with a starting dilution of 1:4. Serial serum dilutions were incubated at 37˚C for 60 min with an equal volume of RSV A2 to provide 100 PFU per 100 µl. The serum-virus mixtures were added in duplicate to monolayers of HEp-2 cells in 96-well flat-bottom tissue culture-treated microtiter plates and incubated at 37˚C with 5% $CO_2$ in DMEM 5% FBS. After 48 h, cells were washed with 100 µl PBS per well and fixed for 20 min at RT with methanol containing 2% $H_2O_2$. Cells were washed twice with 100 µl PBS per well, and biotinylated anti-RSV antibody (goat polyclonal IgG, Bio-Rad) was added for 1 h at RT. Cells were washed twice with 100 µl PBS 1% BSA per well and ExtrAvidin-Peroxidase (Sigma) was added for 30 min at RT. Cells were washed twice with 100 µl PBS 1% BSA per well and DAB substrate (Sigma) was added and left up to 30 min at RT. When the plaques appeared with the desired coloration, the reaction was stopped with PBS. Serial two-fold dilutions of Palivizumab (Synagis), beginning at 100 µg/ml, and virus-free DMEM were used as controls. The plates were scanned with an ImmunoSpot analyzer (Cellular Technology Limited).

## RNA isolation and quantitative PCR

Total RNA was extracted from BAL fluids and nasal swabs using the QiAamp Viral RNA Mini Kit (QIAgen) and Nucleospin RNA Plus Kit (Macherey Nagel) for lung tissues and the BAL cellular fraction according to the manufacturer's protocol. For quantitation of RNA expression levels, quantitative PCR (qPCR) was performed using AgPath-ID One-Step RT-PCR Reagents (ThermoFisher), according to the manufacturer's instructions. The primer and probe sequences used to detect RSV A2 and the housekeeping control 18S RNAs have been previously described: RSV A2 (FW: 5′-GAACTCAGTGTAGGTAGAATGTTTGCA-3′, RV: 5′-TTCAG CTATCATTTTCTCTGCCAAT-3′, Probe: 5′-FAM-TTTGAACCTGTCTGAACATTCCCG GTT-TAMRA-3′) [64]; 18S (FW: 5′-CGCCGCTAGAGGTGAAATTCT-3′, RV: 5′-CATTCT TGGCAAATGCTTTCG-3′, Probe: 5′-FAM-ACCGGCGCAAGACGGACCAGA-TAMRA-3′) [65]. RNA expression levels of RSV and 18S were determined by absolute quantification by the serial dilution of plasmids containing the sequence of interest. For the comparison between RSV A2 and RSV-ON1-H1, the following panRSV primers were used: FW: 5′- TGCTAAGA CYCCCCACCGTAAC-3′, RV: 5′-GGATTTTTGCAGGATTGTTTATGA-3′, Probe: 5′-C5CT 6GC7CT87W7CA-BHQ1-3′ [66]. For the quantification of IFN-β, primers were designed as follow: FW: 5′-TGGTTCTCCTGCTGTGTTTCTC-3′; RV: 5′-CGTTGTTGGAATCGAAGCA A-3′; Probe: 5′-FAM-ACCACAGCTCTTTCCAGGAGCTACA-BHQ1-3′ [67]. For the amplification on ABI Fast 7500 Sequence Detection System (Applied Biosystems), the following

conditions were used: 95˚C 20 sec, 40–45 cycles of the following: 95˚C 3 sec, 60˚C 30 sec, real-time data were analyzed using the SDS software (Applied Biosystems).

### Flow cytometry

The different ovine immune cell subtypes were identified by FCM using an eight-step, seven-color staining protocols. All antibodies used for the procedure, as well as their clones, host, working dilution, and references, are listed in S1 Table. Combination stainings analyzed pDCs, γδ T cells, Tregs, CD4$^+$ and CD8$^+$ T cell subsets. For the acquisitions, 10$^6$ events were accumulated for each sample. The experimental schedule is summarized in S2 Table. For the counts of cell subtypes, we calculated the percentage of total events: ratio (number of events in the gated cell subtype) to (number of all events, excepting cell aggregates and debris). FCM acquisitions were performed on a FACS Canto II (BD Bioscience) using the DIVA software and further analyzed with FlowJo (TreeStar).

### Multiplex immunoassay

For multiplex assay, the fluidic fraction of BALs from mock controls and RSV A2 were collected. Cytokine concentrations were assessed using the MILLIPLEX Ovine Cytokine/Chemo-kine Panel 1 (Life Science Research) according to the manufacturer's protocol and read on a Luminex MAGPIX System including a xPONENT Software version 4.2 software (Luminex).

### Statistical analysis

Statistical analysis was done using the GraphPad Prism 8 software (GraphPad software, La Jolla, CA, USA). Data are presented as box and whisker plots and to determine differences between two groups, non-parametric paired Wilcoxon tests, Mann–Whitney U-tests or one-way *ANOVA* were used as appropriate. Associations were tested using the Spearman rank correlation test. A p value < 0.05 was considered statistically significant.

### Supporting information

**S1 Fig. RSV A2 replication in the ovine model.** (A) Ovine WD-AEC cultures from 3 independent animals were infected with RSV A2 (MOI = 1 PFU/cell). Apical washes were harvested at different time points (24, 48, 72 and 96 h p.i.) and titrated on HEp-2 cells. (B) Nasal swabs from RSV A2-infected neonates were taken daily on a 6 day-duration period. RSV A2 was then quantified by qPCR. Each sample was measured in duplicate. After the initial decrease of RSV A2 titers, a second peak was detected in 5 out of 6 animals, showing the capacity of RSV A2 to replicate *in vivo* in the ovine model.
(TIF)

**S2 Fig. Longitudinal evaluation of peripheral blood over the course of RSV A2 disease.** Frequencies in the peripheral blood of (A) White blood cells (WBC) and (B) polymorphonuclear leukocytes (PMNs) over the course of RSV A2 disease. Data were obtained with VETSCAN HM5 Hematology Analyzer (Abaxis) and graphs were generated with R v.3.4.4 (2018-03-15).
(TIF)

**S3 Fig. Screening for the initial presence of RSV A2-specific neutralizing antibodies in animals euthanized at day 3 or 6 post-infection.** Serums collected in animals prior RSV A2 infection (baseline) were co-incubated with RSV A2 (100 PFU) and applied to HEp-2 cells for 48 hours. The results show that all tested animals were negative. Cell culture medium and

Palivizumab were used as negative and positive controls, respectively.
(TIF)

**S4 Fig. Evaluation of serum RSV A2-specific neutralizing antibodies.** Serums collected in animals prior RSV A2 infection (baseline) or at different time points as indicated, were co-incubated with RSV A2 (100 PFU) and applied to HEp-2 cells for 48 hours. RSV-infected neonates have naturally acquired NAb at day 13–14 p.i. (A), that persist over a 42-day long period (B). Each symbol represents an individual animal (symbols filled with black, healthy adults; transparent symbols, mock neonates; solid color symbols, neonates infected with RSV). Cell culture medium and Palivizumab were used as negative and positive controls, respectively.
(TIF)

**S5 Fig. RSV A2-specific neutralizing activity present in one uninfected animal.** Serums collected in animals at the initial phase of RSV A2 infection (day 3) or at day 42 p.i., were co-incubated with RSV A2 (100 PFU) and applied to HEp-2 cells for 48 hours. The mothers of two RSV-infected sibling presented an RSV-neutralization over the 42-day long period of the experiment. Each symbol represents an individual animal.
(TIF)

**S6 Fig. Absolute cell counts of pDCs and T cell subsets in the bronchoalveolar space upon RSV A2 infection.** (A) Total cellularity measured in the BALs. Cell count and viability was done with Trypan Blue exclusion dye. (B-F) Absolute cell counts of pDCs (B), γδ T cells (C), CD4$^+$ T cells, (D), CD8$^+$ T cells (E) and Tregs (F). Each symbol represents an individual animal (healthy adults, n = 12; mock neonates, n = 3 per time point; neonates infected with RSV, n = 6 per time point). Boxplots indicate median value (center line) and interquartile ranges (box edges), with whiskers extending to the lowest and the highest values. Groups were compared using Mann–Whitney U-tests (A-F). Stars indicate significance levels. $^*$, $p < 0.05$; $^{**}$, $p < 0.01$; $^{***}$, $p < 0.001$. (G) Correlation coefficient (r) obtained in BALs with γδ T cells (% of total events) calculated as a function of pDCs (% of total events), CD4$^+$ T cells, (% of total events), CD8$^+$ T cells (% of total events) and Tregs (% of total events). Absence of negative correlation shows that the γδ T-cell depletion observed in Fig 3A is not related to the expansion of another immune cell subset.
(TIF)

**S7 Fig. FCM gating strategy for immune cells identification.** Example of gating strategy for multiparameter FCM analysis of ovine pDCs (A), γδ T cells (B), CD4$^+$ and CD8$^+$ T cells (C) and Tregs (D).
(TIF)

**S8 Fig. Total cellularity measured in the BALs of animals infected with RSV A2 (adults and neonates) and RSV-ON1-H1 (neonates).** Cell count and viability was done with Trypan Blue exclusion dye. Each symbol represents an individual animal (RSV A2 infected adults, n = 3 per time point; neonates infected with RSV A2, n = 8 per time point; neonates infected with RSV-ON1-H1, n = 4 per time point). Boxplots indicate median value (center line) and interquartile ranges (box edges), with whiskers extending to the lowest and the highest values. Groups were compared using one-way ANOVA followed by Turkey's post hoc test.
(TIF)

**S9 Fig. Distinct cellular immune response in neonates compare to adults following RSV A2 infection.** (A) As in Fig 7G, but with plots integrating all animals. Each symbol represents an individual animal (RSV A2 infected adults, n = 3; Neonates infected with RSV A2, n = 8 per time point). Boxplots indicate median value (center line) and interquartile ranges (box edges),

with whiskers extending to the lowest and the highest values. Groups were compared using one-way ANOVA followed by Turkey's post hoc test. (B) As in Fig 7K and 7O, but with histograms integrating all individuals. Groups were compared using Mann–Whitney U-tests. Stars indicate significance levels. *, p < 0.05.
(TIF)

**S1 Table. List of antibodies validated for their use in the study.**
(DOCX)

**S2 Table. Combination immunostainings used for the identification of cell subsets.**
(DOCX)

## Acknowledgments

We thank Artur Summerfield, Christoph Aebi and Sean R. R. Hall for helpful discussions; Sylvie Python, Aurélie Godel and Nathan Leborgne for technical assistance; Inês Berenguer Veiga for participation in the animal trial; Hans Peter Lüthi, Roman Troxler, Jan Salchli, Daniel Brechbühl and Katarzyna Sliz for animal care. The RSV isolate RSV-ON1-H1 was kindly provided by Sibylle Haid, Martin Wetzke, Gesine Hansen and Thomas Pietschmann.

## Author Contributions

**Conceptualization:** Thomas Démoulins, Nicolas Ruggli, Marco P. Alves.

**Data curation:** Thomas Démoulins, Marco P. Alves.

**Formal analysis:** Thomas Démoulins.

**Funding acquisition:** Marco P. Alves.

**Investigation:** Thomas Démoulins, Melanie Brügger, Beatrice Zumkehr, Blandina I. Oliveira Esteves, Kemal Mehinagic, Amal Fahmi, Loïc Borcard, Adriano Taddeo, Damian Jandrasits, Horst Posthaus, Charaf Benarafa, Nicolas Ruggli, Marco P. Alves.

**Methodology:** Thomas Démoulins, Melanie Brügger, Beatrice Zumkehr, Blandina I. Oliveira Esteves, Kemal Mehinagic, Amal Fahmi, Adriano Taddeo, Damian Jandrasits, Horst Posthaus, Charaf Benarafa, Nicolas Ruggli, Marco P. Alves.

**Project administration:** Marco P. Alves.

**Resources:** Marco P. Alves.

**Software:** Loïc Borcard.

**Supervision:** Marco P. Alves.

**Validation:** Marco P. Alves.

**Writing – original draft:** Thomas Démoulins.

**Writing – review & editing:** Marco P. Alves.

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
