## [Decision Letter · Decision Letter 0]

28 Nov 2020

Dear Dr. Alves,

Thank you very much for submitting your manuscript "The specific features of the developing T cell compartment of the neonatal lung are a determinant of respiratory syncytial virus immunopathogenesis" for consideration at PLOS Pathogens. As with all papers reviewed by the journal, your manuscript was reviewed by members of the editorial board and by several independent reviewers. In light of the reviews (below this email), we would like to invite the resubmission of a significantly-revised version that takes into account the reviewers' comments.

We cannot make any decision about publication until we have seen the revised manuscript and your response to the reviewers' comments. Your revised manuscript is also likely to be sent to reviewers for further evaluation.

Sincerely,

Nicholas W Lukacs

Guest Editor

PLOS Pathogens

Carolina Lopez

Section Editor

PLOS Pathogens

Kasturi Haldar

Editor-in-Chief

PLOS Pathogens

orcid.org/0000-0001-5065-158X

Michael Malim

Editor-in-Chief

PLOS Pathogens

orcid.org/0000-0002-7699-2064

Reviewer's Responses to Questions

**Part I - Summary**

Reviewer #1: The manuscript from Demoulins, et al. utilizes a lamb model of neonatal RSV infection to examine the T cell response to the virus. The advantage of the lamb model is that the lungs of lambs are similar to human lungs and so are a better model than other animal models that are commonly used for understanding RSV infection. The authors infected lambs a week after birth and followed them for 6 weeks. This allowed for data to be obtained at various points during development, a strength of the study. Weaknesses in data presentation including only showing a representative FACS plot and not tabulated data and using Pie charts without showing means and errors. Statistical analysis is also not appropriate for all of the data.

Reviewer #2: The overall objective of this manuscript is to characterize a neonatal lamb model of respiratory syncytial virus (RSV) infection. The rationale for conducting the study is based upon distinct clinical outcomes of RSV infection in young children relative to adults. Although both children and adults can become infected with RSV, severity of symptoms including distal airways inflammation is pronounced during early life. Increased susceptibility to severe RSV disease in young children is thought to be mediated by an immature immune system, but little is known about the developing immunity in the respiratory tract.

To address this gap in knowledge, the authors of the study infected 6 day old neonatal lambs with a human RSV strain (n=6 at each timepoint). The outcome of infection was compared with mock (PBS) infected neonatal lambs (n=3 at each timepoint). Cellular and humeral immune responses were compared with uninfected adults (n=3 at each timepoint). For a portion of the study, the neonatal cellular immune response to RSV infection was compared in a small cohort of infected adults (n=3 at 6 and 14 d.p.i.). Two different strains of RSV virus were used, including the ATCC RSV-A2 strain as well as the RSV-ON1-H1 strain isolated by the authors from an infected child.

Strengths of the study include the importance of understanding early life lung immunity, particularly in the context of respiratory viruses. There are several important limitations of the study findings. The primary read out for virus is PCR, however this method does not establish replication. Because TLR3 activation does not require infectious virus, a treatment group with UV inactivated virus would have been an additional important control. The authors do find a number of modest differences in lavage/tissue cell populations, which could be strengthened by statistics.

Reviewer #3: The study examines an important area i.e. immunity and disease after RSV infection in neonates. The study and model are being sold by the authors as the gold standard and one that properly recapitulates human disease. If this is going to be the case, then the authors need to discuss how the model recapitulates (or does not) the diversity in the response in human infants and children. Not all children have severe disease. Where does the model fail to recapitulate the response of human children undergoing their 1st RSV infection, age 1… 6 months old, etc. Issues of passive Ab and others are not examined. What are the strengths and weaknesses of the model. Percentages of cells rather than cell numbers are examined throughout. Percentages do not always reflect the response as they are dependent on other cell populations remaining similar and total cell count being equal. The data herein suggest those other cell populations are not remaining the same. Thus interpretation of the percentages is difficult. For many datasets only a single representative sample is provided. It is unclear how representative these single samples are.

**Part II – Major Issues: Key Experiments Required for Acceptance**

Reviewer #1: 1. Statistical analysis is described in methods as non-parametric tests between two groups. However, some of the data looks like analysis of variance or the non-parametric equivalent would be more appropriate (Figs 5c and E for example).

2. In the discussion the authors suggest (line294-6) that pDC might have defects in signaling by RSV or deficiency in cytokine production. Do you have any data on type I interferon production by pDCs?

3. It is curious that the authors did not discuss the possibility the ILCs could have a role in the TH2 bias they see early after infection. This should at least be part of the discussion if they did not examine them directly.

Reviewer #2: 1. It would be helpful to provide more details for the overall experimental design to understand the different animal groups. The schematic in Figure 1 is acceptable but additional information on procedures conducted at each time point presented in an organized fashion (e.g. table that accompanies the figure) would improve the manuscript.

2. Figure legends could use more detail. It was often difficult to understand how the data were collected and analyzed (both main and supplement figures)

3. As discussed in the summary, it would be important for the authors to discuss the limitations of the study with regards to viral replication. This text can be included in the discussion.

Reviewer #3: 1) In Figure 1, why is there a positive viral titer in the mock group? Is there transfer of the virus in cohoused animals that complicates this analysis? What are the samples from in Fig 1C? Is this only the neonates? The RSV+ type I pneumocytes and macrophages should be indicated in these images. How were the macrophages and type I pneumocytes identified? In addition to these provided images the results of several samples should be quantified. Figure 1E, so were all the healthy adults screened to be RSV negative in order to be included in that study group. It is strange that the mother in question that passively transferred Ab to the neonate appears to have been positive. What was the seropositivity of the adult animals in the facility? This also introduces an additional important parameter to the study/model for it to be translational. How does the current model reflect (or not) human infants which may have passive antibody from their mothers. It appears the response in the 2 animals that had Ab may result in lower titers at day 13/14. Figure 1G, the figure this reviewer had to examine does not contain any data for days 3, 6, and 42 only data at day 14. This needs to be fixed.

2) Why do mock levels of gd T cells also drop in the BAL in Figure 2A. In addition to the representative samples shown, data in Figure 2C-E for all animals examined should be shown in graphs as in Fig 2A and the results analyzed by statical analysis. Why does the IL-17 level in the mock increase at day 42 in Figure 2E? If IFNg is being analyzed the percentages should be provided.

3) Data in Fig 3 A/B should be shown as cell numbers rather the percentages, as the percentages can vary based on changes in other cell populations. 100 T cells among 1000 total cells and 100 T cells among 10,000 cells is still only 100 T cells in the tissue. In Figs 2 and S4/5 the authors argue for other cell populations to be changed in the RSV infected neonates. Thus, the percentages here are very difficult to interpret as these other populations are changeing. The text comments on IFNg not being different but no numbers for IFNg+ events are provided on the plots to allow the reader to assess this. Likewise numbers should be provided in Fig 3D/F due to the fact the number of T cell in each group may not be equivalent. Finally, it has become clear in many experimental models that many of the T cells particularly in the uninfected individuals are in the circulation and not in the lung airways or interstitium. Thus many groups have begun to label cells in the circulation with Ab in order to accurately measure only those cells in the lung interstitium and airways. The important T cells found within the interstitium, and contribute to disease control and immunopathology, are not examined herein.

**Part III – Minor Issues: Editorial and Data Presentation Modifications**

Reviewer #1: 1. Data are interesting, but some of it needs to be in a different format to make it clear whether there are actually differences between groups. The authors chose to put what they call significant data on pDCs in supplemental figure 5 rather than show the data in the body of the paper. This is not helpful and the data should be in the main body of the paper.

2. Figure 2 shows FACS plots without showing all of the data in graph form. This could be in supplemental data since the authors state it is not statistically significant. Contour plots in Fig 5 F,I, and L show what are probably percentages in lower right quadrants with an asterisk associated with them. There is no indication from the figure legends what these numbers are. Are they means? If means, what are the standard errors and how many individuals were used for the calculation? This is an example of using graphs to show these data. Several figures have Pie charts in them which are unhelpful from the standpoint that the reader can’t see standard error and not statistical analysis is indicated. I you want to keep the pie charts for the visual they provide then the means and standard errors should be in the supplemental data or in the text.

3. Symbols used in figure 5 (purple) appeared there suddenly without any explanation. Fig 6B does not indicate which viral strain goes with which set of data.

Reviewer #2: 1. What is the basis for the dose of virus? It is also unclear which animals received the A2 strain versus H1 strain. Was the H1 strain exclusively used for Figure 6? What was the rationale for selecting the two strains?

2. Is there evidence of viral replication? This is critical to confirm the immune outcomes and pathology are due to infection versus an inflammatory response to the inactivated virus. PCR measures are informative but are not sufficient to confirm infection; plaque assays are needed. As an alternative, comparative analysis using mock treatment with UV inactivated virus would confirm infection as opposed to TLR mediated inflammation.

3. It is surprising that no differences in blood PMN were detected giving the apparent detection of inflammation in the lung. Are PMN readily visible in the alveolar compartment of RSV infected animals?

4. Were cell counts and basic cell phenotyping conducted on BAL samples (e.g. cytospin of lavage cells)? This would provide quantitative measures of key cell types responding to the RSV, including macrophages, neutrophils, eosinophils, and lymphocytes.

5. The authors show an increase in the BAL pDC population with RSV but the numbers are very low. While it would not be surprising to find low pDC frequency in BAL, the overwhelming presence of macrophages in BAL samples may create autofluorescence background issues by FACS. Autofluorescence can be further exacerbated when using reagents that are not directly conjugated with fluorochromes. What controls were used to validate the FACS panel for pDC?

6. It is difficult to interpret Figure 2B. Are the authors inferring that gamma delta T cells are highest when there is more overall cell death due to RSV? Why would gamma delta T cells decline?

7. Lines 176-177, the authors state “an unexpected reduction of CD4+ T-cell recruitment was seen as soon as 14 days p.i.”. It this reduction significantly different? The change appears modest at best.

8. For the study described in Figure 5, did the adults and neonates receive the same dose/volume of virus? If so, what is the difference in lung volume between the two ages and could the lack of inflammation in adults be explained by dilution effect?

Reviewer #3: 1) The uniqueness of this work is being oversold. There have been other studies on T cell immunity and neonatal responses. Thus, the authors selling of the work as “unprecedented” (line 92) and other similar descriptions in the manuscript seems as though the authors are unaware of prior studies some of which they cite.

2) The figure design in many places is not clear, well labeled or adequately explained in the legends/text.

3) How are the responses in the 2 animals with passive antibody, or other animals with passive Ab, altered in the subsequent figures and analysis?

4) Data in S4 and S5 should be included in the paper and not be supplemental.

5) Fig 4 A, please see comments about the need to distinguish cells in the lungs vs circulation above.

6) What is the basis for the statement on line 235-6 that the response is tightly regulated?

7) Overall day 0 or controls should be shown for all groups in Fig 5. Again all data should be shown for figures such as Fig 5D/F/G/I/J/L/M not just representative samples. Statistical analysis should be undertaken.

8) Cell numbers not just percentages should be shown in Fig 6 (see issues with only examining the % that are discussed above).

PLOS authors have the option to publish the peer review history of their article (what does this mean?). If published, this will include your full peer review and any attached files.

Reviewer #1: No

Reviewer #2: No

Reviewer #3: No
---

## [Decision Letter · Decision Letter 1]

21 Mar 2021

Dear Dr. Alves,

Thank you very much for submitting your manuscript "The specific features of the developing T cell compartment of the neonatal lung are a determinant of respiratory syncytial virus immunopathogenesis" for consideration at PLOS Pathogens. As with all papers reviewed by the journal, your manuscript was reviewed by members of the editorial board and by several independent reviewers. The reviewers appreciated the attention to an important topic. Based on the reviews, we are likely to accept this manuscript for publication, providing that you modify the manuscript according to the review recommendations.

Please address the comments by Reviewers within the text of the manuscript. No new experiments are needed but a careful adjustment to some of the result and concluding statements should be made to clarify the modeling and interpretation of the data.

Sincerely,

Nicholas W Lukacs

Guest Editor

PLOS Pathogens

Carolina Lopez

Section Editor

PLOS Pathogens

Kasturi Haldar

Editor-in-Chief

PLOS Pathogens

orcid.org/0000-0001-5065-158X

Michael Malim

Editor-in-Chief

PLOS Pathogens

orcid.org/0000-0002-7699-2064

Please address the comments by Reviewers within the text of the manuscript. No new experiments are needed but a careful adjustment to some of the result and concluding statements should be made to clarify the modeling and interpretation of the data.

Reviewer Comments (if any, and for reference):

Reviewer's Responses to Questions

**Part I - Summary**

Reviewer #1: The authors have extensively addressed the concerns of the reviewers including adding more data and cleaning up the presentation and statistical analysis of data.

Reviewer #2: The overall objective of this revised manuscript is to investigate the pulmonary T cell response to respiratory syncytial virus (RSV) infection using a neonatal lamb model. RSV is a highly relevant infection in young children that can result in hospitalization and prevention is limited to prophylaxis. The authors conduct immune cell profiling of normal lungs and compare with lungs following viral infection, with the rationale that the T cell response may elicit a distinct signature with RSV. Neonatal responses are compared with adult animal responses. In addition, two different variants of RSV are tested to assess for differences in ability to promote local immune responses. Strengths of the manuscript include the overall clinical relevance of understanding the distinct pulmonary immune response to RSV within pediatric populations and the use of a large animal model. As written, there are a number of points within the manuscript that appear to overstate the outcomes of each experiment.

**Part II – Major Issues: Key Experiments Required for Acceptance**

Reviewer #1: None.

Reviewer #2: (No Response)

**Part III – Minor Issues: Editorial and Data Presentation Modifications**

Reviewer #1: Line 265 should indicate the strain of virus being used is a clinical strain rather than referring the reader to the next section.

Reviewer #2: 1. Lines 67-69 Upon stimulation neonatal T cells tend to differentiate into regulatory T cells (Tregs), to facilitate self-tolerance to developing organs

This sentence needs to be rephrased. As written, it suggests that most neonatal T cells are Tregs. While it is true that neonates have higher frequencies of Tregs, this is not the majority of the T cell population. In the human fetus, there are indeed a higher frequency of Tregs (about 15%) but infants and adults have about 5% in the CD4+ cell population.

2. For Figure 1B, there is no description of what the data on the left represent. Mock? Adults?

3. How are the data in Figure 1G quantified? Are there replicates?

4. Lines 192-193 Finally, we demonstrated the inability of neonatal gamma delta cells to produce detectable levels of IL-17 (Figure 3F, G).

This statement is inconsistent with the figure. IL-17 is detectable at day 42 in the neonates.

5. The authors make the statement that “neonatal RSV A2 infection induces a fast and massive Tc2 influx in the bronchoalveolar space”. However, there does not appear to be a difference in total cell numbers (Figure S8) It is possible that populations may shift over time, such that macrophage numbers are reduced in response to the virus.

6. For Figure 7G, what does WC1 represent?

7. Lines 281-282 From these results, we could ascertain that neonates lack a mature and functional gamma delta T cell

This statement is not reflective of the reported data. Gamma delta cells are detected in the neonate lung lavage but appear to decline following viral infection and therefore produce less IL-17.

8. Lines 283-285 The percentage of IL-4-producing cells was very low…(Figure 7I, J).

This statement is inconsistent with the data shown in Figure 7I and J.

9. Lines 290-292 From these results, it became clear that the inaptitude of neonates to mount an effector T cell response is tentatively substituted by a fast and massive Th2/Tc2 cell recruitment.

This statement is not supported by the data. What is the evidence that the effector T cell response is ineffective, particularly in light of comparable generation of neutralizing antibodies between neonates and adults (Figure 6D)

10. Line 292 …we failed to detect any T reg recruitment

This statement is not supported by the data (Figure 5A)

11. For Figure 8, authors report that the T cell response is a correlate of severity of lung inflammation in association with RSV strains. There appears to be trend toward increased viral replication with RSV A2, but there are no statistical differences A2 and ON1-H1. It’s difficult to make a conclusive statement about the magnitude of the T cell responses from these data because of the small sample size, unless T cell numbers can be quantitatively correlated with inflammation (e.g. tissue neutrophil numbers).

PLOS authors have the option to publish the peer review history of their article (what does this mean?). If published, this will include your full peer review and any attached files.

Reviewer #1: No

Reviewer #2: No

Figure Files:

Data Requirements:

Reproducibility:

References:

---

## [Editor Report · Decision Letter 2]

5 Apr 2021

Dear Dr. Alves,

We are pleased to inform you that your manuscript 'The specific features of the developing T cell compartment of the neonatal lung are a determinant of respiratory syncytial virus immunopathogenesis' has been provisionally accepted for publication in PLOS Pathogens.

Best regards,

Nicholas W Lukacs

Guest Editor

PLOS Pathogens

Carolina Lopez

Section Editor

PLOS Pathogens

Kasturi Haldar

Editor-in-Chief

PLOS Pathogens

orcid.org/0000-0001-5065-158X

Michael Malim

Editor-in-Chief

PLOS Pathogens

orcid.org/0000-0002-7699-2064
---

## [Editor Report · Acceptance letter]

23 Apr 2021

Dear Dr. Alves,

We are delighted to inform you that your manuscript, "The specific features of the developing T cell compartment of the neonatal lung are a determinant of respiratory syncytial virus immunopathogenesis," has been formally accepted for publication in PLOS Pathogens.

Best regards,

Kasturi Haldar

Editor-in-Chief

PLOS Pathogens

orcid.org/0000-0001-5065-158X

Michael Malim

Editor-in-Chief

PLOS Pathogens

orcid.org/0000-0002-7699-2064